# A Hox-TALE regulatory circuit for neural crest patterning is conserved across vertebrates

Hugo J. Parker [1], Bony De Kumar[1], Stephen A. Green[2], Karin D. Prummel[3], Christopher Hess[3], Charles K. Kaufman[4], Christian Mosimann [3], Leanne M. Wiedemann [1,5], Marianne E. Bronner[2] & Robb Krumlauf[1,6]

In jawed vertebrates (gnathostomes), *Hox* genes play an important role in patterning head and jaw formation, but mechanisms coupling *Hox* genes to neural crest (NC) are unknown. Here we use cross-species regulatory comparisons between gnathostomes and lamprey, a jawless extant vertebrate, to investigate conserved ancestral mechanisms regulating *Hox2* genes in NC. Gnathostome *Hoxa2* and *Hoxb2* NC enhancers mediate equivalent NC expression in lamprey and gnathostomes, revealing ancient conservation of *Hox* upstream regulatory components in NC. In characterizing a lamprey *hoxα2* NC/hindbrain enhancer, we identify essential Meis, Pbx, and Hox binding sites that are functionally conserved within *Hoxa2/Hoxb2* NC enhancers. This suggests that the lamprey *hoxα2* enhancer retains ancestral activity and that *Hoxa2/Hoxb2* NC enhancers are ancient paralogues, which diverged in hindbrain and NC activities. This identifies an ancestral mechanism for *Hox2* NC regulation involving a Hox-TALE regulatory circuit, potentiated by inputs from Meis and Pbx proteins and Hox auto-/cross-regulatory interactions.

[1] Stowers Institute for Medical Research, Kansas City, MO 64110, USA. [2] Division of Biology and Biological Engineering, California Institute of Technology, Pasadena, CA 91125, USA. [3] Institute of Molecular Life Sciences, University of Zurich, 8057 Zürich, Switzerland. [4] Division of Medical Oncology, Washington University in Saint Louis, St. Louis, MO 63110, USA. [5] Department of Pathology and Laboratory Medicine, Kansas University Medical Center, Kansas City, KS 66160, USA. [6] Department of Anatomy and Cell Biology, Kansas University Medical Center, Kansas City, KS 66160, USA. Correspondence and requests for materials should be addressed to M.E.B. (email: mbronner@caltech.edu) or to R.K. (email: rek@stowers.org)

Neural crest (NC) cells are a migratory and multipotent cell type, representing a defining characteristic of vertebrates[1–3]. The cranial NC emerges from the mid/hindbrain region and contributes to cartilage and bone of the pharynx. Gnathostomes (jawed vertebrates) exhibit nested *Hox* gene expression domains (*Hox* codes) along the anteroposterior (AP) extent of both the neural tube and adjacent NC. However, little is known regarding shared versus independent regulation of *Hox* expression in these two tissues. In NC of the pharyngeal arches (PAs), the NC *Hox* code confers positional identity to each arch[4]. There is also evidence for an earlier role of *Hox* genes in regional generation of NC from the hindbrain[4,5]. Emergence of NC and its underlying *Hox* code played an important role in craniofacial evolution, giving rise to unique structures of the head and neck, including the jaws and hyoid apparatus used in feeding and respiration. Genome analyses together with gene expression and functional studies have described a common framework for a gene regulatory network (GRN) that drives NC induction, specification, and migration across vertebrates, and some regulatory components are present in non-vertebrate chordates[1,6,7]. However, *Hox* genes have not yet been integrated within the current formulation of the NC GRN. This is in part because the mechanisms regulating *Hox* expression in NC are relatively unclear compared to the current knowledge of *Hox* regulation in hindbrain segmentation[4]. Hence, despite the established functional roles of *Hox* genes in cranial NC[4,8], whether the *Hox* code is coupled to this conserved NC GRN and if so how, is unknown. It is unclear whether *Hox* networks that control axial patterning and generation of NC are integrated within or are working in parallel, independently from the NC GRN.

*Hox2* genes are critical components of the cranial NC *Hox* code and provide an important avenue for addressing this question. Experimental alterations in expression of gnathostome *Hox2* paralogues (*Hoxa2* and *Hoxb2*) lead to homeotic transformations. In mice, ectopic *Hoxa2* suppresses jaw formation[9], while *Hoxa2* mutants exhibit mirror image jaw duplication, indicating a role as a selector gene in NC AP identity[4,8]. Regulatory analyses in mouse, chick, and zebrafish have identified evolutionarily conserved enhancers in the *Hoxa2* and the *Hoxb2* genes that mediate their expression in NC and hindbrain segments (rhombomeres (r)) (Fig. 1a, b)[4,10–13]. Analyses of mouse *Hoxa2* and *Hoxb2* NC enhancers provide conflicting mechanisms for NC expression of *Hox* genes. A single 5′-flanking enhancer drives *Hoxb2* in r4 and its NC[11], supporting a model for shared regulation in these tissues (Fig. 1b). In contrast, independent exonic/intronic and 5′-flanking enhancers regulate *Hoxa2* expression in r4[12] versus r4-derived NC[14] (PA2), suggesting the evolution of distinct regulatory mechanisms between the hindbrain and NC (Fig. 1a). Characterisation of *cis*-elements required for the activity of these enhancers has provided some insight into their underlying regulatory mechanisms (Fig. 1a, b). Both the *Hoxb2* r4/NC and the *Hoxa2* exonic/intronic r4 enhancers depend upon the combined inputs from Meis and Pbx-Hox binding sites for their activities[11,15–17]. However, the inputs into the independent *Hoxa2* NC enhancer are largely unknown, except for a binding site for transcription factor AP2-α (Tfap2α) in the mouse enhancer that is not conserved across gnathostome species[10,12]. Taken together, these analyses reveal differences between *Hoxa2* and *Hoxb2* in the enhancers and regulatory mechanisms underlying their expression in r4 and NC. Hence, in mouse, each gene supports a different model with respect to shared versus independent regulation in the hindbrain and NC. Given that *Hoxa2* and *Hoxb2* are paralogous genes, this raises two questions: what is the evolutionary relationship between their NC enhancers and which of these mechanisms is ancestral?

The sea lamprey offers an opportunity to address conserved ancestral mechanisms of *Hox2* NC regulation in vertebrates. The extant jawless vertebrates (cyclostomes), lamprey and hagfish, represent a sister group to gnathostomes, so comparisons between these lineages can provide insights into the evolution of gene regulatory programs in vertebrates[18–20]. Such comparisons can reveal aspects of vertebrate biology that were present in the common ancestor of extant vertebrates and which have been conserved in each lineage. These studies can also identify features that differ between each lineage, which could represent divergence from the ancestral vertebrate state in either/both lineages. While comparisons between cyclostomes and gnathostomes may reveal ancestral features of vertebrate *hox2* regulation in the NC, little is known about the enhancers and regulatory factors underlying *Hox* NC expression in cyclostomes. Here, we employ cross-species regulatory comparisons between lamprey and gnathostomes to isolate and characterize NC enhancers and investigate ancestral regulation of *Hox2* in vertebrates.

## Results

**Expression of *hox2* genes in lamprey cranial NC.** In two species of lamprey, sea lamprey (*Petromyzon marinus*) and Arctic lamprey (*Lethenteron camtschaticum*), only two *Hox2* paralogues (*hoxα2* and *hoxδ2*) have been identified, but their orthology to gnathostome *Hoxa2/Hoxb2* remains unresolved (Fig. 2a)[21–23]. The sea lamprey has six *Hox* clusters, compared to the four clusters inferred in ancestral gnathostomes[22]. A recent reconstruction based on comparisons of gene order at the chromosomal level between vertebrate species supports a model in which the ancestor of cyclostomes and gnathostomes also had four *Hox* clusters[22], suggesting that the two additional *Hox* clusters in lamprey arose from duplication event/s in the lamprey/cyclostome lineage. To investigate the duplication history of lamprey *Hox* clusters, previous work employed a chromosome-wide analysis of genomic synteny (duplicate gene retention) between lamprey *Hox*-bearing chromosomes[22]. These pairwise comparisons indicated that the chromosomes bearing the *hoxα* and *hoxδ* clusters display a significantly closer relationship to each other than to any other *Hox*-bearing chromosomes. Similarly, the chromosomes carrying the *hoxβ* and *hoxε* clusters also show a significant enrichment for shared paralogues with each other. This suggests that these pairs of chromosomes derive from duplication event/s that occurred more recently than the duplication events that gave rise to the other *Hox*-bearing chromosomes[22]. Phylogenetic analyses reveal *hoxβ* and *hoxε* paralogues consistently clustering in protein trees[21,22], while *hoxα* and *hoxδ* paralogues do not show any clear patterns of clustering. This may be due to the limitations of phylogenetic analyses in resolving the relative timing of ancient duplication events[22]. Thus, the evidence from synteny analysis leads us to infer that *hoxα2* and *hoxδ2* genes are paralogues that arose from duplication in the lamprey/cyclostome lineage.

In exploring *hox* gene NC regulation in sea lamprey embryos, our time-course analysis of *hox1–3* expression revealed nested domains in the pharynx from stage (st) 23, reminiscent of those in Arctic lamprey[24,25] and in gnathostomes[4,8] (Fig. 2b–e). *hoxα2* is expressed in the NC posterior to PA1 and in the hindbrain posterior to r1, similar to gnathostome *Hoxa2*. In contrast, *hoxδ2* is expressed in r3/r5, notochord, and in posterior pharyngeal endoderm and mesenchyme, but not at high levels in NC (Fig. 2b, c). If *hoxα/δ* clusters arose by duplication in the lamprey/cyclostome lineage, as supported by synteny analysis, this suggests that *hoxδ2* expression diverged after this duplication. Given the similarity in NC expression of *hoxα2* to gnathostome *Hoxa2*, we focused our comparative regulatory analysis on mechanisms mediating lamprey *hoxα2* expression in NC.

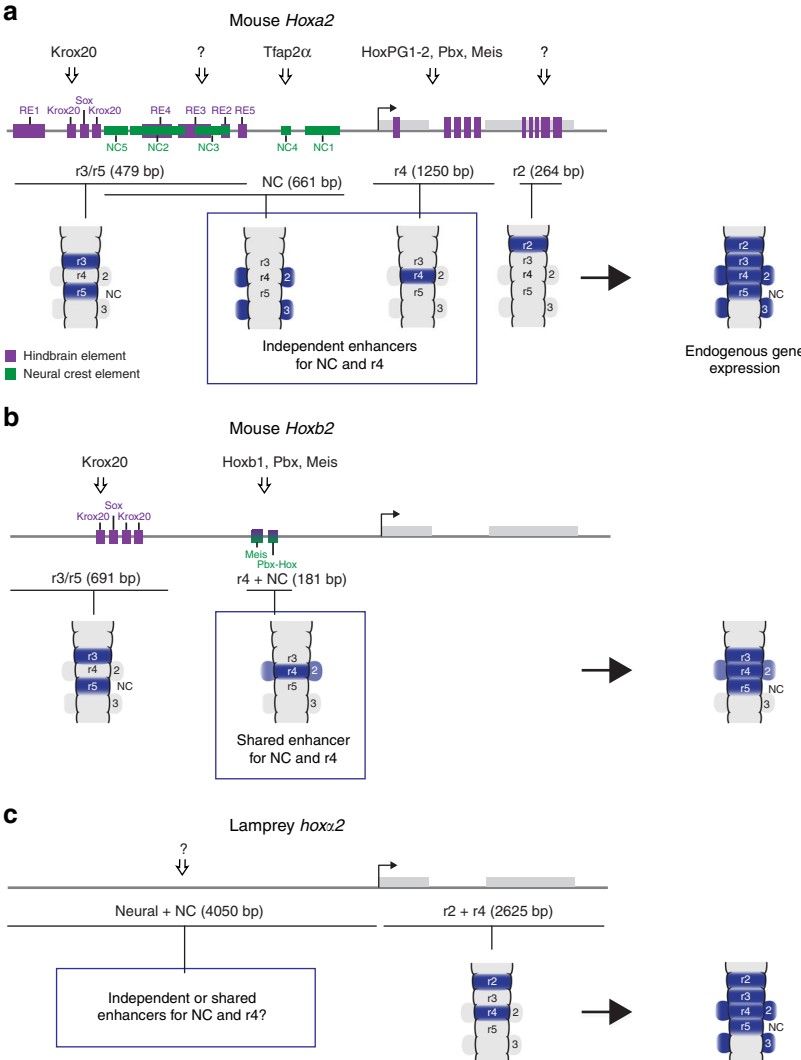

**Fig. 1** Characterized *Hox2* enhancers regulating neural crest (NC) and rhombomeric expression from mouse and lamprey. Schematic diagrams depicting the known enhancers regulating mouse *Hoxa2* (**a**) and *Hoxb2* (**b**), and lamprey *hoxα2* (**c**), in rhombomeres (r) and NC. For each locus, the gene exons are represented by grey boxes and the transcriptional start site by an arrow. Enhancers are marked as black lines below the loci, with their activity domains illustrated by blue shading in schematic dorsal views of the hindbrain (r2–5) and pharyngeal arches (2–3). For the mouse loci, characterised *cis*-elements contributing to enhancer function are depicted as coloured boxes: hindbrain elements in purple and NC elements in green (not drawn to scale). Known direct inputs from transcription factors into these *cis*-elements are depicted by arrows, with unknown inputs shown as question marks. *Hoxa2* is regulated in r4 and r4-derived NC (PA2) by independent enhancers (**a**). *Hoxb2* expression in r4 and NC is mediated by a single enhancer, through *cis*-elements bound by Meis and Pbx-Hox factors (**b**). Since these elements have dual hindbrain/NC activity they are depicted in both purple and green. Genomic regions from the lamprey *hoxα2* locus have enhancer activity, with an r2/r4 enhancer positioned within the exons and intron (**c**). The *hoxα2-hoxα3* intergenic region drives reporter expression in the hindbrain and NC. However, it is not known whether this is through independent or shared NC/hindbrain enhancers, specific *cis*-elements have not been identified, and the relationship of this region to the gnathostome *Hoxa2* and *Hoxb2* enhancers is unclear

**Gnathostome *Hox2* NC enhancers function in lamprey.** To explore conservation of the regulatory network upstream of *Hox2* in the NC, we sought to perform cross-species enhancer analysis by testing gnathostome NC enhancers for activity in lamprey embryos. As part of this approach, we wanted a way to globally monitor the NC in vivo. In zebrafish, the *crestin* promoter/enhancer element is a highly specific tool for monitoring NC development, as it is active from pre- to post-migratory stages across multiple axial levels, making it a good candidate for marking NC in lamprey[26]. In lamprey transgenic reporter assays, the *crestin* element mediates spatiotemporal expression in NC similar to its activity in zebrafish, and its activity is sensitive to perturbation of the same transcription factor binding sites (Supplementary Figure. 1a–d; Fig. 3b; Supplementary Table 1).

Initial green fluorescent protein (GFP) expression in pre-migratory NC was observed at st21 and maintained as NC delaminated and migrated ventro-laterally to populate the pharynx (Supplementary Figure 1b–c). Frontal sections at st24 revealed that GFP transcripts are present in the NC-derived pharyngeal mesenchyme (Fig. 3d). These data demonstrate that NC cells can be labelled and visualized in vivo by a reporter assay in the developing lamprey and suggest that the *crestin* element is interpreted in lamprey by conserved upstream components of an ancestral NC GRN. This serves as a proof of concept for interspecies analysis of NC enhancers.

Next, we investigated whether upstream regulatory inputs required for *Hoxa2* expression in gnathostome NC are present in lamprey using homologous *Hoxa2* NC enhancers from three

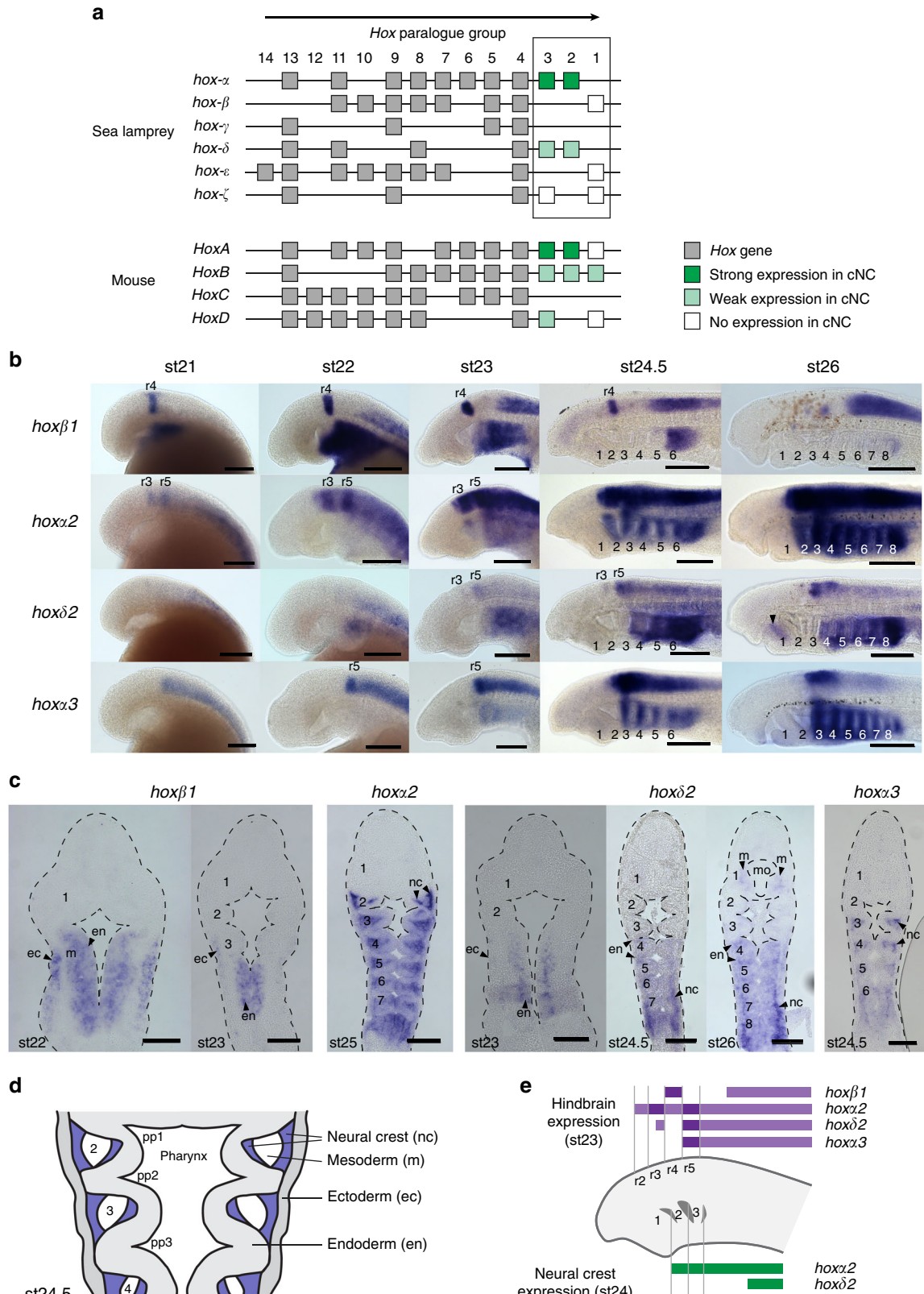

gnathostomes (zebrafish (zf), fugu (f), mouse (m)) (Fig. 3a). In both lamprey and zebrafish embryos, all three gnathostome enhancers mediated GFP reporter expression in the developing pharynx posterior to PA1 (Fig. 3b; Supplementary Table 2; Supplementary Figure 2). Frontal sections of transient transgenic lamprey embryos confirmed that GFP transcripts were

prominently expressed in NC-derived pharyngeal mesenchyme equivalent to *crestin* reporter expression domains and endogenous *hoxa2* (Fig. 3d).

In gnathostomes, NC *Hoxa2* expression is regulated by 5′-flanking elements (*NC1–5*) that partially overlap those of a separate r3/r5 enhancer (*RE1-5, Krox20, Sox*) (Figs. 1a, 3a)[10,27].

**Fig. 2** Embryonic time course showing expression of *hox* genes in the lamprey hindbrain and cranial neural crest (NC). **a** Genomic organization of *Hox* genes in lamprey and mouse. Boxes represent *Hox* genes, which are organized into paralogue groups based on their sequence. The arrow above the clusters denotes the direction of *Hox* gene transcription. Lamprey *hox* genes from paralogue groups 1–3 were examined for NC expression in this study and their expression in cranial NC is denoted by green/white shading. **b** Lateral views of lamprey embryos from stages (st)21 to 26, showing *hox* gene expression domains in the developing head. Pharyngeal arches are numbered and rhombomere-specific domains (r) indicated. The arrowhead marks weak *hoxδ2* expression in mandibular mesoderm at st26. **c** Frontal sections through lamprey embryos showing *hox* gene expression domains within the developing pharynx. Pharyngeal arches are numbered. **d** Schematic of a frontal section through the lamprey st24.5 embryonic pharynx with tissue domains annotated; NC domains are shaded in blue. Scale bars: 200 μm (**b**); 100 μm (**c**). **e** Schematic depicting *hox* expression in the lamprey hindbrain and NC at st23 and st24. ec, ectoderm; en, endoderm; m, mesoderm; mo, mouth; nc, neural crest; r, rhombomere; st, stage

Of these, the *NC3* element is the most highly conserved: global sequence alignment using Multi-LAGAN[28] identified sequence conservation of *NC3* extending to sharks (Fig. 3a). In previous work, two 15 bp deletions within *NC3* were each found to abolish NC reporter expression in mouse[10]. To determine whether the same *cis*-elements are required for NC activity of *Hoxa2(m)* in lamprey, we generated two variants with these deletions in *NC3*. While hindbrain activity persisted, these reporters exhibited severely diminished NC activity in zebrafish and lamprey (Fig. 3a, c; Supplementary Figure 3a, b; Supplementary Table 3). Analogously, activity of a gnathostome *Hoxb2* NC enhancer (*hoxb2a (zf)*) also depended upon the same regulatory sites in lamprey as in zebrafish (Supplementary Figure 3c, d). Since *Hoxa2(m)* and *hoxb2a(zf)* NC enhancers require the same *cis*-motifs across distant species, this suggests that an ancestral GRN upstream of *Hox2* in NC patterning has been retained in lamprey and gnathostome lineages.

**Conservation of binding sites in a lamprey *hoxα2* NC enhancer**. Gnathostome *Hoxa2* and lamprey *hoxα2* share similarity in their *cis*-regulatory architecture for rhombomeric hindbrain expression, as r2 and r4 enhancers are embedded in conserved locations (Fig. 1a, c)[13,18]. However, our global sequence alignment failed to reveal conservation of NC regulatory elements (*NC1–5*) 5′ of the lamprey *hox2* genes (Fig. 3a). Hence, we manually searched for short sequence motifs within an enhancer (*hoxα2 −4 kb*) in the 5′-flanking region of *hoxα2* that recapitulates endogenous *hoxα2* expression in the hindbrain (r3–r5), NC posterior to PA1 and somites (Fig. 4a–e; Supplementary Figure 4; Supplementary Table 2). In mouse *Hoxa2*, the most highly conserved elements required for NC expression are *NC3* and part of *NC2*, while those necessary for r3/r5 activity are *Krox20*, *Sox*, *RE2*, and *RE3* sites. Surveying the lamprey enhancer, we identified short matching sequences for *Krox20*, *Sox*, and *NC3* (Fig. 4f; Supplementary Figure 5). To address whether a smaller region containing these sites retains enhancer activity, we cloned a 1530 bp region encompassing the conserved sites with ~500 bp on each side (*hoxα2 elementA*) and demonstrated that it mediates reporter expression equivalent to *hoxα2 −4 kb* in lamprey (Fig. 4g, j). To test if the conserved sites are required for tissue-specific enhancer activity, we assayed variants of *hoxα2 −4 kb* in which these sites were mutated (Fig. 4f, h–j). Deleting *Krox20-Sox* sites (Δ*Krox20*) resulted in the loss of r3/r5 expression, but maintenance in r4 and NC (Fig. 4f, h). In contrast, deleting the conserved sites within *NC3* (Δ*NC3*) caused loss of all rhombomeric and NC activities, but somitic expression is retained (Fig. 4f, i–j).

Inspection of the conserved sites between the lamprey *hoxα2* and gnathostome *Hoxa2* enhancers revealed that they match closely to consensus transcription factor binding site motifs for factors involved in early hindbrain and NC patterning. In addition to the previously characterised Krox20 sites, we identified three short blocks of conserved sequence that correspond to consensus binding motifs for Sox, Meis, and Pbx-Hox factors (Fig. 4f). These motifs each fall within regions

functionally required for enhancer activity in gnathostomes and lamprey, notably *NC3* (Fig. 4a, f, h–j, Supplementary Figure 3a–b), suggesting that Meis, Pbx, and Hox factors may provide conserved and essential inputs into these enhancers. Thus, lamprey *hoxα2* and gnathostome *Hoxa2* appear to be regulated in the hindbrain and NC through conserved transcription factor binding sites retained during vertebrate evolution. This provides further support for an ancestral GRN upstream of *Hox2* in NC patterning that has been retained in lamprey and gnathostomes.

**Hoxa2 and Hoxb2 NC enhancers are divergent paralogues**. The identification of conserved Meis, Pbx, and Hox binding motifs in the lamprey *hoxα2* and gnathostome *Hoxa2* enhancers is significant as equivalent sites are present and functionally required in the mouse *Hoxb2* and zebrafish *hoxb2a* enhancers (Fig. 1b; Supplementary Figure 3c). To explore whether these are homologous sites and to interrogate common and diverged features of the *hoxα2*, *Hoxa2*, and *Hoxb2* enhancers, we used the *Krox20* sites, implicated in r3/r5 expression, as an anchor to align the enhancer sequences. This revealed striking conservation of the sequence and order of *Krox20*, *Sox*, *Meis*, and *Pbx-Hox* sites between these enhancers, with relatively low sequence conservation within the intervening regions (Fig. 5a; Supplementary Figure 6). Based on these conserved sites and relative positions, we infer that *Hoxa2* and *Hoxb2* NC enhancers are ancient paralogues, derived from an ancestral vertebrate *Hox2* enhancer that contained these sites. These paralogous enhancers appear to have diverged in gnathostomes, such that the mouse *Hoxa2* enhancer is inactive in r4 but expressed in r4-derived NC, while the *Hoxb2* enhancer is active in both r4 and its NC (Fig. 5a)[11]. The lamprey *hoxα2* NC enhancer exhibits the combined activity of both mouse *Hoxa2* and *Hoxb2* enhancers, suggesting that it may reflect the ancestral state. We searched upstream of lamprey *hoxδ2*, finding conservation of *Krox20* and *Sox* sites, but no *Meis* or *Pbx-Hox* sites (Fig. 5b). This suggests that the ancestral sites for r4/NC enhancer activity were lost upstream of *hoxδ2*, consistent with the expression of *hoxδ2* in r3/r5 but not in r4 and r4-derived NC (Fig. 2b, c).

**Lamprey *meisC* and *hoxα2* are similarly expressed in NC**. Identification of Meis and Pbx-Hox consensus binding motifs in the *hox2* enhancers of lamprey and gnathostomes implies that these factors may play conserved roles in regulating vertebrate *Hox2* genes in the hindbrain and NC. Meis, Prep, and Pbx are members of the TALE (Three-Amino-Acid-Loop-Extension) homeodomain family[29] that have diverse roles in patterning tissues, including hindbrain and NC[30–33]. Additionally, they can act as cofactors for other transcription factors, including Hox proteins[34]. To initially explore whether Meis factors are linked with regulation of NC in lamprey, we characterized expression of four lamprey *meis* genes. We found that *meisC* shows early NC expression with a spatiotemporal pattern similar to *hoxα2*

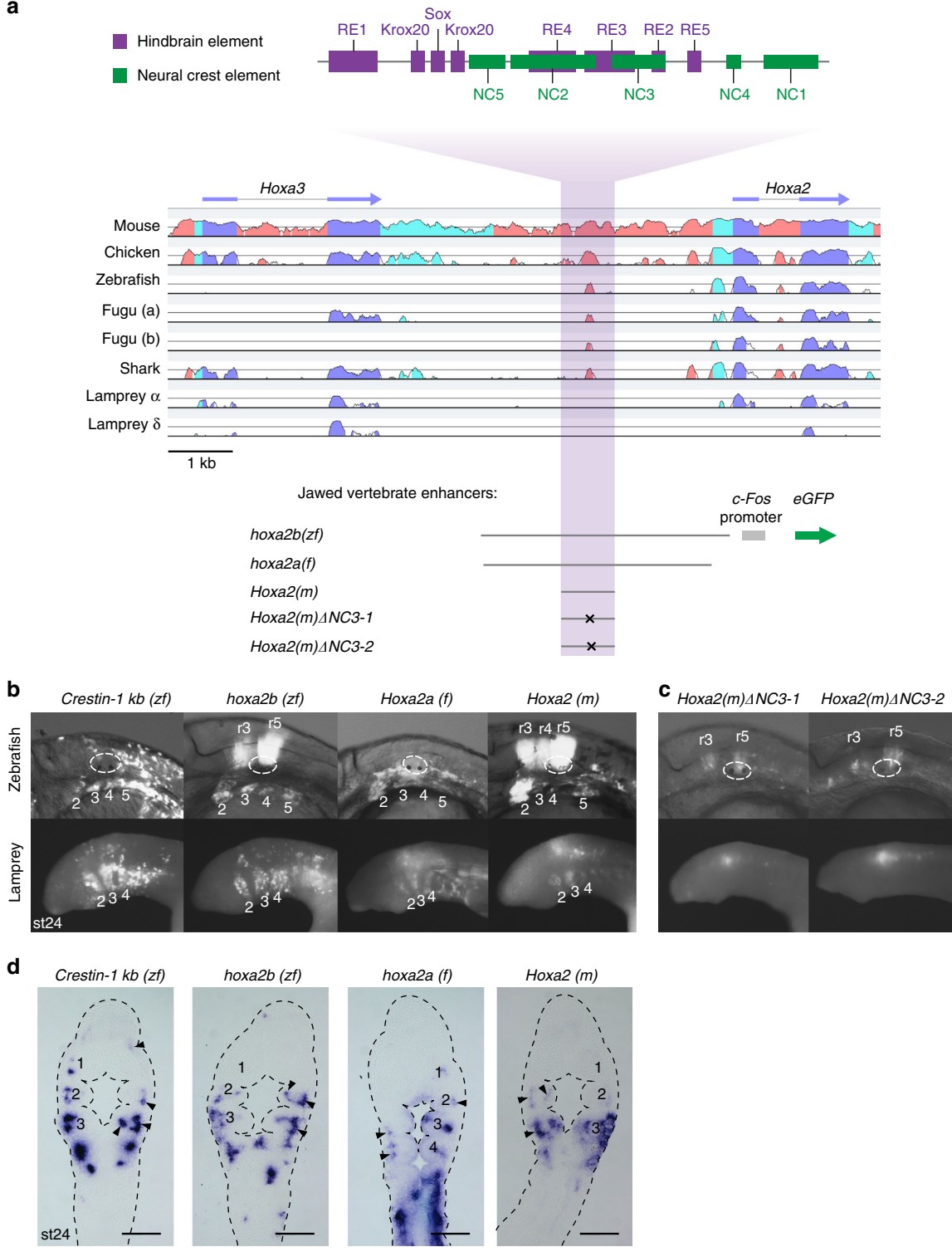

**Fig. 3** Conserved activity of gnathostome *Hoxa2* neural crest (NC) enhancers in zebrafish and lamprey. **a** Sequence alignment of gnathostome *Hoxa2-Hoxa3* and lamprey *hox2-hox3* gene loci against the human locus. Conserved non-coding sequences (pink), untranslated regions (UTRs) (cyan) and coding sequences (blue) are highlighted. The relative locations of the mouse hindbrain and NC *cis*-elements (top) are shown. Gnathostome *Hoxa2* enhancers used for cross-species reporter analysis are detailed below the alignment. Letters within parenthesis indicate species of origin of the enhancer: zf, zebrafish; f, fugu; m, mouse. **b**, **c** Green fluorescent protein (GFP) reporter expression in zebrafish and lamprey embryos (lateral views), mediated by wild-type (**b**) and mutated (**c**) gnathostome NC enhancers. For zebrafish, the otic vesicle is circled and GFP expression in rhombomeres (r) and pharyngeal arches (2–5) indicated. Lamprey pharyngeal arches are labelled (2–4). GFP-expressing embryos shown are representative of the expression potential of the reporter construct in each case, as inferred from screening many (typically more than 100) injected embryos. Supplementary Table 2 provides the number of embryos and details of specific expression for all constructs in lamprey. Injection statistics for the transient transgenic zebrafish embryos shown in **c** are given in Supplementary Table 3. **d** Frontal sections through the transient transgenic lamprey embryos shown in Fig. 3b, with GFP transcripts detected by in situ hybridisation, revealing expression in NC-derived mesenchyme (arrowheads) in the pharyngeal arches (numbered). Scale bars: 100 μm

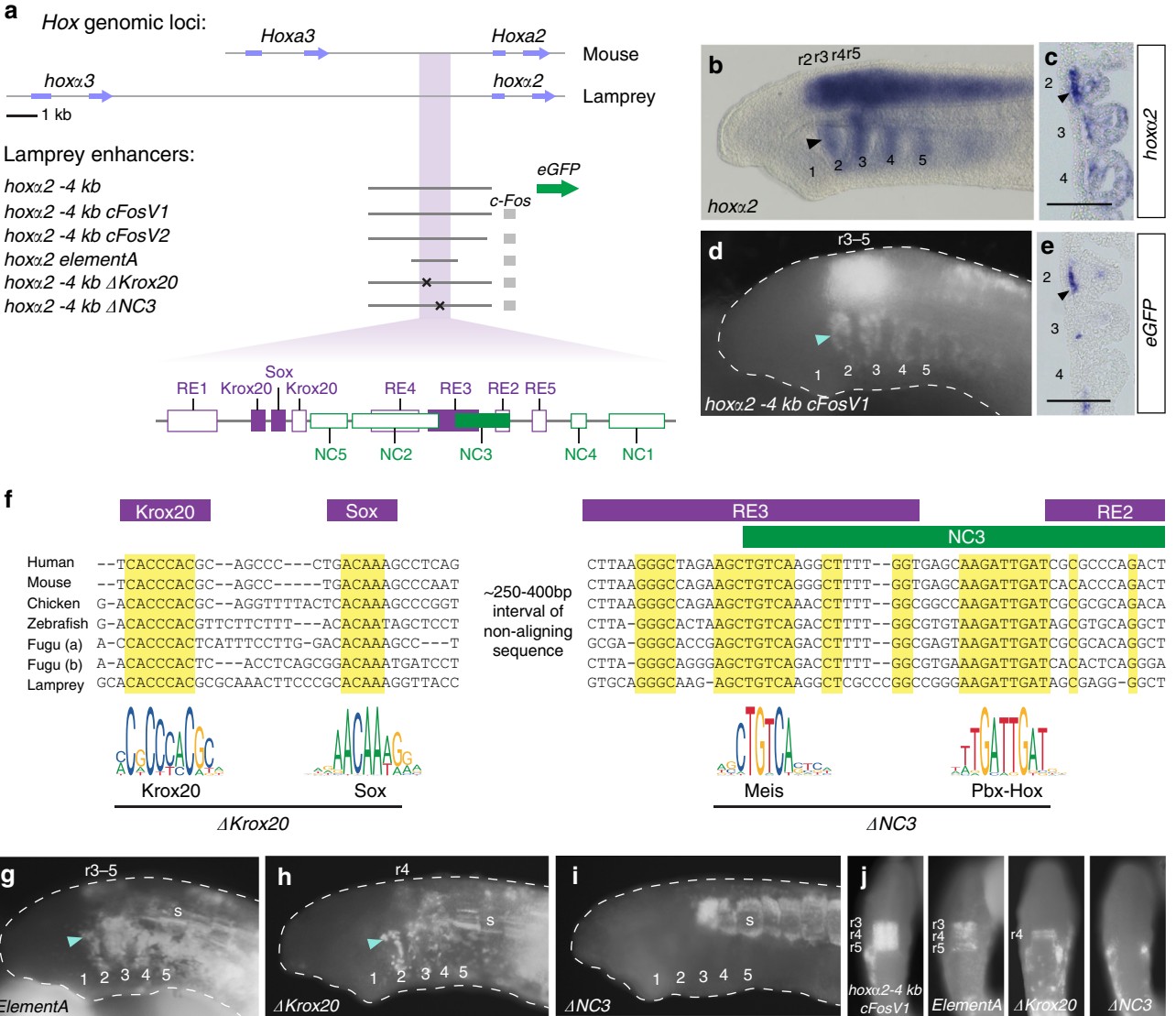

**Fig. 4** Characterization of a lamprey *hoxa2* neural crest (NC)/hindbrain enhancer. **a** The mouse *Hoxa2-Hoxa3* genomic region and its equivalent from the lamprey *hoxα* cluster are shown, with *Hox* gene exons annotated (blue arrows). *hoxα2* upstream regions assayed for reporter activity in this study, with or without the *c-Fos* minimal promoter, are shown. **b–e** Lateral views (**b**, **d**) and frontal sections (**c**, **e**) of st24.5 lamprey embryos, comparing endogenous expression of *hoxα2* (**b**, **c**) to GFP reporter expression mediated by *hoxα2 −4 kb* (**d**, **e**). Pharyngeal arches are numbered and rhombomeric expression detailed. Arrowheads point to PA2 NC expression. **f** Multiple sequence alignment of the *Hoxa2* NC enhancer from gnathostomes with the lamprey *hoxα2* enhancer, showing conserved sites (yellow). The positions of characterized mouse *cis*-elements (*Krox20*, *Sox*, *RE2-3*, *NC3*) are marked above the alignment. The enhancer schematic (**a**) shows the position of these elements within the assayed *hoxα2* upstream regions, with conserved (shaded boxes) or divergent (empty boxes) *cis*-elements highlighted. Consensus binding motifs from the JASPAR database[76] for Krox20[77], Sox11[78], Meis1[79], and Pbx-Hox[80] factors are shown below the alignment, as well as sequences deleted in *hoxα2 −4 kb ΔKrox20* and *ΔNC3* variants. The non-aligning interval between these conserved regions is ~250–400 bp and varies in length between species. Supplementary Figure 5 contains the full alignment. **g–j** Lateral (**g-i**) and dorsal (**j**) views of st24.5 lamprey embryos showing GFP reporter expression driven by the enhancers detailed in **a**. Pharyngeal arches are numbered, with expression in rhombomeres (r) and somites (s) annotated. GFP-expressing embryos shown are representative of the expression potential of the reporter construct in each case, as inferred from screening many (typically more than 100) injected embryos. Supplementary Table 2 provides the number of embryos and details of specific expression for all constructs in lamprey

(Fig. 6a–d). This expression data and the conserved consensus Meis binding sites are consistent with the notion that *meis* genes may have played an ancestral role in regulating *Hox2* in NC during vertebrate evolution.

**Occupancy of TALE and Hox proteins on *Hox2* enhancers.** Hoxb1, Pbx, and Meis bind to sites within the mouse *Hoxb2* hindbrain/NC enhancer that are required for its activity[11,17]. The presence of homologous sites within NC3 of the mouse *Hoxa2*

enhancer suggests that such factors may also regulate its activity in the hindbrain and NC. This is significant because previous characterisation of this NC enhancer did not uncover factors that bind to NC3 nor provide insight into its underlying regulatory mechanism[10,35]. The divergent activities of the *Hoxa2* and *Hoxb2* enhancers, particularly in r4, suggest that there may be differences in their interactions with upstream regulatory factors despite the presence of homologous binding sites. This led us to investigate binding properties of Meis, Pbx, and Hox proteins on these enhancers.

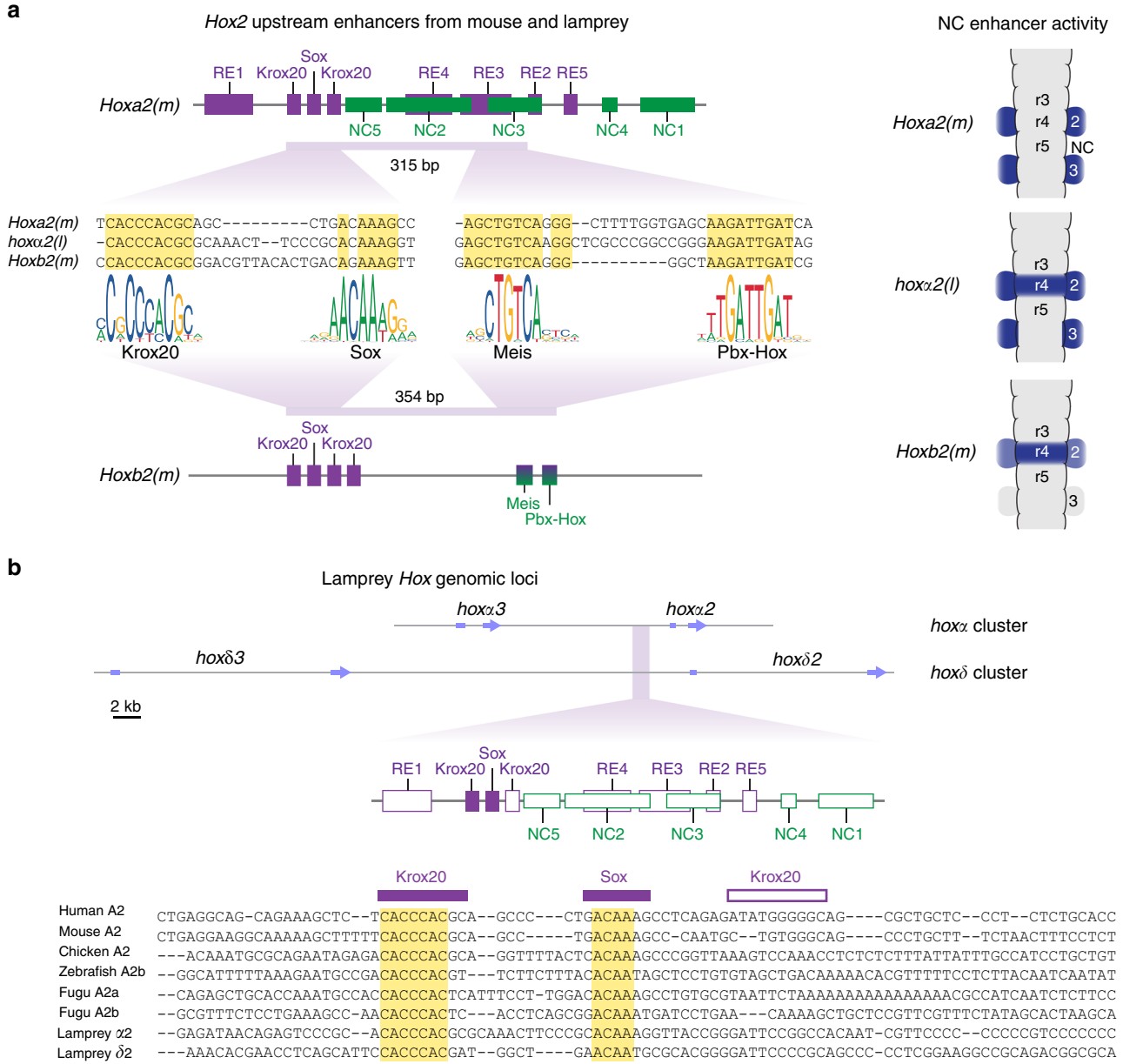

**Fig. 5** *Hoxa2* and *Hoxb2* neural crest (NC) enhancers are ancient paralogues and the lamprey *hoxα2* enhancer appears to reflect the ancestral state. **a** Sequence alignment of mouse (m) *Hoxa2* and *Hoxb2* NC enhancers with that of lamprey (l) *hoxα2*, revealing short conserved sequence blocks (yellow). Corresponding consensus binding motifs for Krox20, Sox, Meis, and Pbx-Hox factors are shown below the alignment. These conserved sequences map to characterized *cis*-elements required for hindbrain (purple) or NC (green) activity in the mouse *Hoxa2* (above) and *Hoxb2* (below) enhancers. The 315 and 354 bp refer to the precise distances between the 5′ end of the Krox20 site and the 3′ end of the *Pbx-Hox* site of the mouse *Hoxa2* and *Hoxb2* enhancers, respectively. The activity of each NC enhancer in the hindbrain and NC is shown in schematic dorsal views. This activity differs between each enhancer, with *hoxα2(l)* showing the combined output of *Hoxa2(m)* and *Hoxb2(m)*. **b** Multiple sequence alignment of gnathostome *Hoxa2* NC enhancers with a homologous region upstream of lamprey *hoxδ2*. The lamprey *hoxα2-hoxα3* and *hoxδ2-hoxδ3* genomic loci are depicted, with *hox* gene exons annotated (blue arrows). The multiple sequence alignment reveals conservation of a *Krox20* and a *Sox* site upstream of *hoxδ2* (yellow shading in alignment), but other *cis*-elements, including *NC3*, are not conserved in sequence. This is depicted in the enhancer schematic, which details the conserved (shaded boxes) and divergent (empty boxes) *cis*-elements upstream of *hoxδ2*

For insight into NC, we harnessed published genome-wide binding data for Meis, Pbx, and Hoxa2 in PA2 of mouse embryos[36,37]. Focusing on the NC enhancers of *Hoxa2 (NC3)* and *Hoxb2 (HRE)*, we observed similar binding profiles for these factors over each of the enhancers, consistent with their regulatory activity in NC. There is enrichment for occupancy of Meis and Pbx, in keeping with a role for these TALE factors in controlling *Hox2* NC activity (Fig. 7a, b). It is interesting

that Hoxa2 also binds to these enhancers in PA2, suggesting that there may be auto- and cross-regulatory inputs from Hox2 proteins into the NC *Hox* code. This is analogous to the important roles for auto- and cross-regulatory circuits in regulating *Hox* expression in other tissues, including hindbrain rhombomeres[11,13,15]. Since Pbx and Meis can act as Hox co-factors[38], they may interact with Hoxa2 on these NC enhancers. Together with the presence of essential Meis motifs

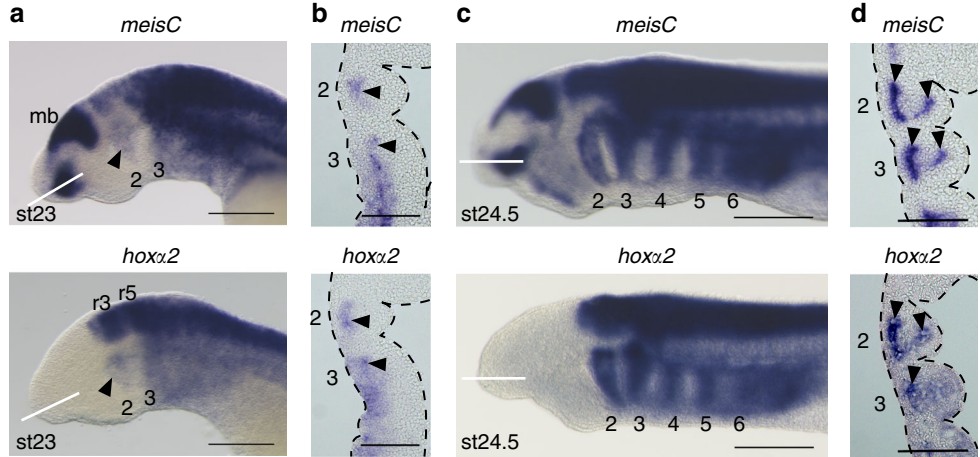

**Fig. 6** Endogenous expression of *meisC* in neural crest (NC) overlaps with that of *hoxα2* in lamprey embryos. **a** Lateral views (**a**, **c**) and frontal sections (**b**, **d**) are shown for embryos at st23 (**a**, **b**) and st24.5 (**c**, **d**). White lines in **a** and **c** denote planes of sections in **b** and **d**. Pharyngeal arches are numbered, arrows denote expression in NC. Scale bars: 200 μm (**a**, **c**); 100 μm (**b**, **d**). mb, mid-brain

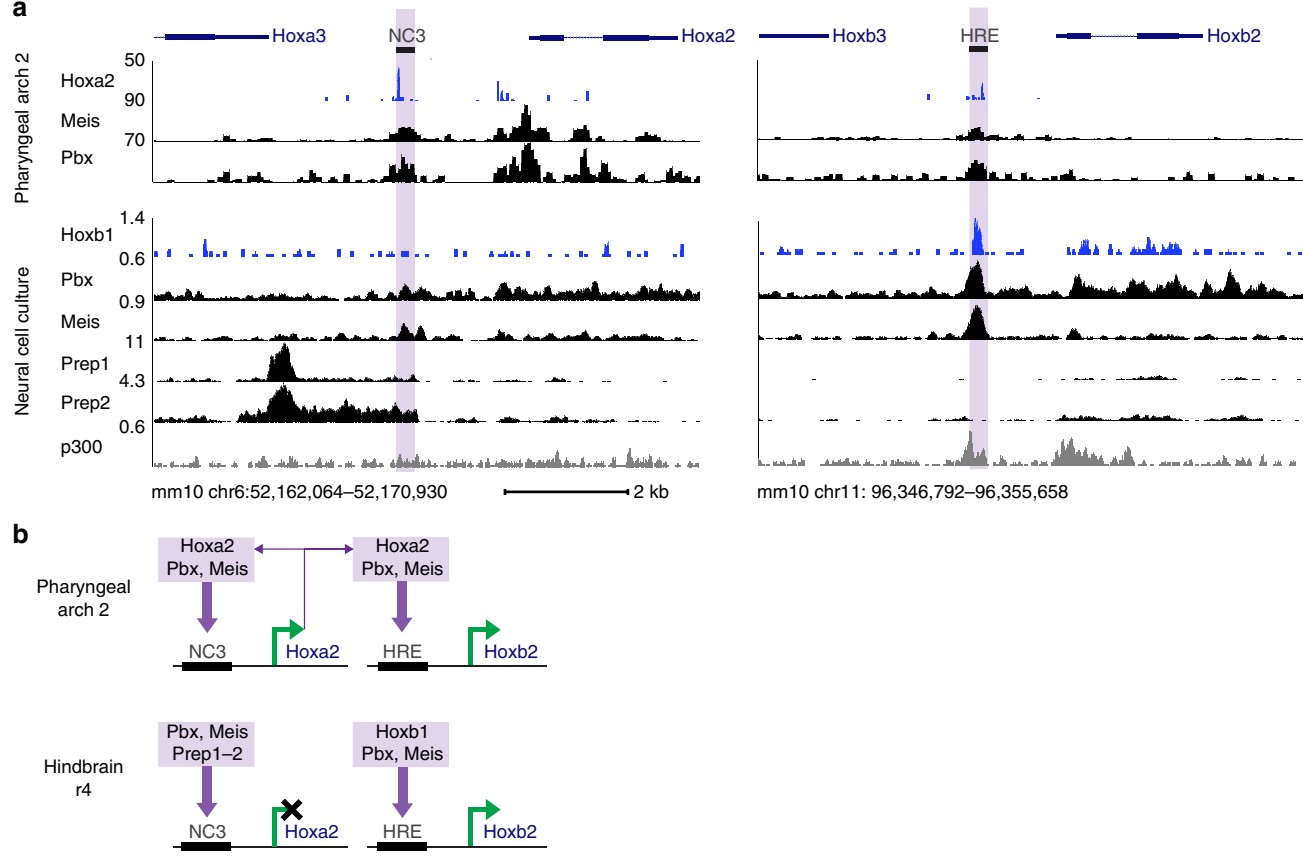

**Fig. 7** *Hoxa2* and *Hoxb2* enhancers exhibit differential TALE (Three-Amino-Acid-Loop-Extension) and Hox binding correlating with their tissue-specific activities. **a** DNA-binding profiles for Hox, TALE, and p300 factors in neural cell culture and pharyngeal arch 2 tissues at the mouse *Hoxa2* (*NC3*) and *Hoxb2* (*HRE*) neural crest (NC) enhancers (highlighted in purple). Genes are annotated (top) and are transcribed from left-to-right. **b** Summary diagram of characterized differential regulatory inputs (purple arrows) from Hox and TALE factors (inferred from **a**) into the mouse *Hoxa2* and *Hoxb2* NC enhancers in pharyngeal arch 2 (NC) and hindbrain r4 in vivo. Activation or inactivation of transcription is depicted by green arrows or a black cross, respectively. Purple arrows from the *Hoxa2* gene indicate auto-/cross-regulation

and bipartite Pbx-Hox sites, this raises the possibility of both *Hox*-dependent and -independent inputs of Pbx and Meis into NC *Hox* expression.

With respect to the developing hindbrain, chromatin immunoprecipitation-sequencing (ChIP-seq) approaches are not feasible for individual rhombomeres due to the small number of cells and limiting amounts of embryonic material. In addition, there is no suitable anti-Hoxb1 antibody for ChIP-seq experiments. To circumvent these challenges, we generated a mouse embryonic stem (ES) cell line (KH2) carrying an inducible

locus-specific insertion of *Hoxb1* marked with Flag epitopes and used it in combination with programmed differentiation of ES cells into neural fates. This enabled us to use anti-Flag antibodies for Hoxb1 ChIP-seq and to obtain sufficient cell populations for a comparative series of genome-wide binding experiments. This cell culture system has previously been shown to exhibit global changes of gene expression that are similar to early in vivo phases of neural development, including the sequential activation of hindbrain-expressed *Hox* genes and their cofactors (such as *Meis2*)[39,40]. We have previously applied this system to investigate the genome-wide binding properties of Hoxa1 and TALE proteins in neural cells, uncovering in vivo regulatory interactions relevant to hindbrain patterning[39–42].

Using this same approach for neural cells, we found similarities and significant differences in binding patterns of Hox and TALE proteins between the paralogous *Hox2* enhancers (Fig. 7a, b). The *Hoxb2 (HRE)* enhancer shows robust binding of Hoxb1, Pbx, and Meis, plus prominent p300 recruitment, consistent with their established in vivo role in mediating *Hoxb2* expression in r4[11,17]. In contrast, the *Hoxa2 (NC3)* enhancer lacks discernable binding of Hoxb1, has reduced levels of Pbx and Meis occupancy, and displays differential binding patterns of Prep1 and Prep2. These properties, in combination with absence of p300, directly correlate with its lack of activity in r4. These differences in TALE and Hoxb1 binding and r4 activity presumably reflect sequence divergence or differences in epigenetic states between the two enhancers. Studies on Pbx-Hox protein binding have shown that small sequence variations within the canonical Pbx-Hox bipartite binding sites influence the selectivity for specific Hox proteins[38]. However, sequence comparisons of the *Hoxb2*, *Hoxa2*, and *hoxα2* enhancers show that the core consensus *Meis* and *Pbx-Hox* sites are identical (Figs. 4f, 5a; Supplementary Figure 6). In contrast, the sequences around these sites differ considerably between enhancers: for example, sequences immediately 5′ of the *Meis* site are shared between *Hoxb2* and *hoxα2* but not *Hoxa2* (Supplementary Figure 6). While these differences in neighbouring sequences may have arisen by sequence drift and be functionally neutral, an intriguing alternative is that they may have functional significance in modulating binding to the conserved motifs and impacting r4 activity. Taken together, ChIP-seq data link TALE (Pbx and Meis) and Hox proteins to regulation of *Hox2* genes in both the hindbrain and NC.

## Discussion

Here, we have investigated the ancestral regulation of *Hox2* genes in the NC and hindbrain of vertebrates, using interspecies *cis*-regulatory comparisons between gnathostomes and lamprey. We demonstrated that gnathostome *Hoxa2* and *Hoxb2* NC enhancers are capable of driving equivalent NC expression in lamprey as they do in gnathostomes and that a homologous enhancer is present upstream of lamprey *hoxα2*. Sequence comparisons and regulatory analysis revealed conserved Meis, Pbx, and Hox binding sites between gnathostome *Hoxa2*/*Hoxb2* and lamprey *hoxα2* NC enhancers that are required for their activity. The lamprey *hoxα2* NC enhancer appears to have retained ancestral activity in both NC and hindbrain, while the paralogous *Hoxa2* and *Hoxb2* NC enhancers have differentially partitioned NC and hindbrain activities in the gnathostome lineage. Regulatory divergence has also occurred between lamprey *Hox2* paralogues, with *hoxδ2* appearing to have lost the regulatory sites for r4/NC enhancer activity. This suggests that a regulatory circuit with input from TALE and Hox proteins was an important component of the GRN for *Hox2*-dependent NC patterning in ancestral vertebrates that has been maintained during evolution (Fig. 8a). TALE factors are part of an ancient patterning system[38,43] that

may have multiple roles in coupling *Hox* expression to the core NC GRN. These findings raise a number of interesting points and avenues for further investigation.

At the mechanistic level, little is known with respect to shared versus independent inputs that govern axial patterning in the hindbrain and NC. This is because the mechanisms regulating *Hox* expression in the NC are relatively unclear compared to the current knowledge of rhombomeric *Hox* regulation[4]. Analyses of mouse *Hoxa2* and *Hoxb2* NC enhancers provided conflicting mechanisms for NC expression of *Hox* genes. *Hoxa2* supported evidence for independent enhancers mediating expression in r4 and r4-derived NC, since the NC enhancer is not active in r4 and a separate intronic/exonic enhancer drives r4 expression[10,15]. In contrast, *Hoxb2* uses common elements to control r4 and NC expression, suggesting similar or shared regulatory requirements in these tissues[11]. Our analyses resolve this paradox, providing evidence that the *Hoxa2*/*Hoxb2* NC enhancers each retain conserved Meis, Pbx, and Hox binding sites, which are deployed in slightly different ways in mediating tissue-specific activities. *hoxα2* is the only lamprey *hox2* paralogue expressed in PA2 NC (Fig. 2b, c) and we uncovered the presence of homologous functional motifs (Meis, Pbx, and Hox) in an upstream enhancer with activity in r4 and NC. Based on sequence conservation, position, and regulatory activity, we infer that this enhancer is homologous to gnathostome *Hoxa2*/*Hoxb2* NC enhancers. Sequence comparisons suggest that lamprey *hoxδ2* has diverged and is missing these NC motifs but retains Krox20 and Sox sites, consistent with its expression in r3/r5. Since the lamprey *hoxα2* NC enhancer exhibits the combined activity of the *Hoxa2* and *Hoxb2* enhancers, we consider it likely that it reflects the ancestral state. Thus, we suggest that *Hox2* was ancestrally regulated in r4 and NC by a shared enhancer through inputs by Meis, Pbx, and Hox (Fig. 8b). Alternatively, if the ancestral NC enhancer did not have the r4 activity, then gnathostome *Hoxb2* and the lamprey *hoxα2* NC enhancers independently evolved the ability to mediate expression in r4.

The loss of r4 activity from the *Hoxa2* NC enhancer may have been mitigated by the existence of a second r4 enhancer, located in the exon/intronic region. We have previously shown that a region containing the lamprey *hoxα2* exon1, intron, and exon2 was also capable of driving reporter expression in r4 in lamprey embryos[18]. This implies that the rhombomeric activity of this genomic region is ancestral to extant vertebrates. Hence, in lamprey *hoxα2* there are two r4 enhancers, which may have partially redundant/shadow activities. It is notable that while both mediate expression in r4 in lamprey, only the upstream enhancer drives expression in NC, which indicates that there is something unique about the upstream enhancer that helps to potentiate its activity in both rhombomeres and NC.

The close proximity and partial overlap between the r3/r5 and r4/NC enhancers in *hoxα2*, in a manner analogous to its gnathostome counterparts (Fig. 1), suggests that some components of these *cis*-elements may be required for expression in both tissues. Our experiments with the lamprey *hoxα2* enhancer showed that deletion of the *Meis* and *Pbx-Hox* sites not only led to the loss of r4 and NC expression but also to the loss of r3/r5 expression (Fig. 4j). This implies that Meis and/or Pbx-Hox factors are also involved in regulating the expression of *hoxα2* in r3/r5. In this regard, the mouse *Hoxa2* NC enhancer partially overlaps elements required for r3/r5 activity: deletion of the *Meis* site in NC3 (ΔNC3_1) causes the loss of NC activity and also reduces r3/r5 expression when tested in mouse[10,44] and zebrafish (Fig. 3c; Supplementary Figure 3a–b). Thus, the *Meis* site contributes to both r3/r5 and NC activities of the *Hoxa2* enhancer. Deletion of the *Pbx-Hox* site (ΔNC3_2) removes NC expression but does not influence r3/r5 activity in mouse[10], which may reflect the control

**a** Evolution of the neural crest *Hox code* in vertebrates

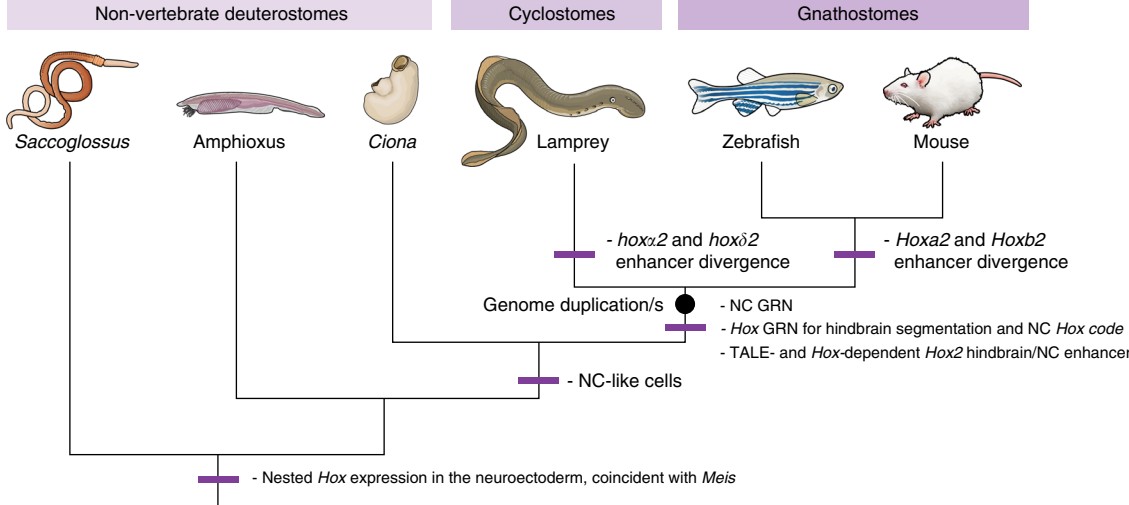

**b** Divergence of *Hox2* hindbrain/NC enhancers

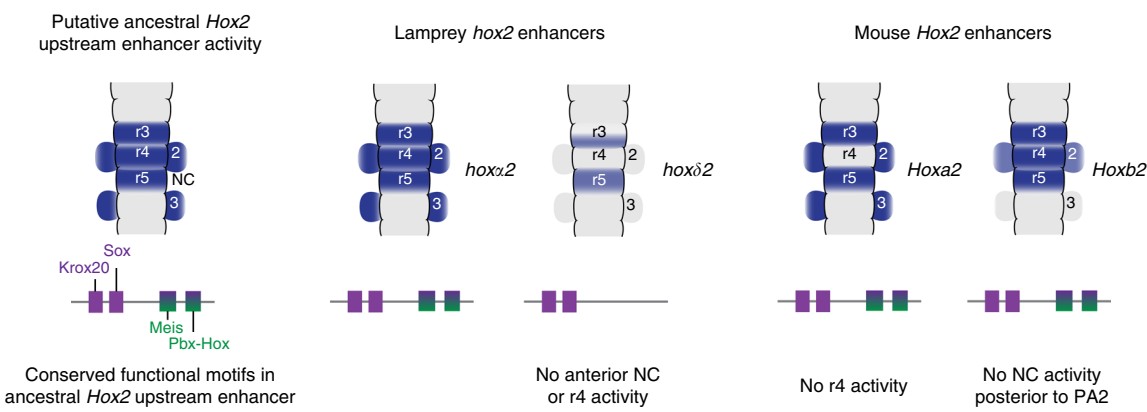

**Fig. 8** Evolutionary model for regulation of *Hox2* genes in vertebrates. **a** A model for evolution of the neural crest (NC) *Hox* code, based on our data. **a** NC gene regulatory network (GRN) and NC *Hox* -code evolved in ancestral vertebrates and are conserved between cyclostomes and gnathostomes. In ancestral vertebrates, *Hox2* NC expression was regulated by TALE (Three-Amino-Acid-Loop-Extension) and Hox factors, through a putative ancestral enhancer with shared NC and hindbrain activities. **b** A model for the divergence of lamprey and mouse *Hox2* NC/hindbrain enhancers. Enhancer activity domains are depicted in blue in schematic dorsal views of the hindbrain and pharyngeal arches. Conserved functional motifs (Krox20, Sox, Meis, Pbx-Hox) present upstream of lamprey and mouse *Hox2* genes are shown. Lamprey and mouse enhancers show divergent activities. Comparison between expression domains and conserved motifs leads us to suggest that a putative ancestral vertebrate *Hox2* enhancer contained *cis*-elements for r3/r5 expression (*Krox20*, *Sox*) and r4/NC expression (*Meis*, *Pbx-Hox*). These scenarios assume that duplication events that gave rise to four *Hox* clusters in early vertebrates occurred prior to the cyclostome/gnathostome split, as the most parsimonious explanation[22,81]. However, it is also possible that independent genome duplication events may have occurred in cyclostome and gnathostome lineages (see Holland and Ocampo Daza[82] for a recent discussion)

of r3/r5 activity being Hox-independent. For *Hoxb2*, r3/r5 activity appears to be independent of the *Meis* and *Pbx-Hox* sites[17,45], suggesting that *Hoxa2* and *Hoxb2* enhancers have also diverged in their degree of dependence on *Meis* sites for r3/r5 activity.

Grafting experiments in gnathostome embryos have revealed roles for both maintenance of neural tube *Hox* expression (pre-patterning) and plasticity in shaping the NC *Hox* code[46–49]. Initial AP *Hox* patterning in the neural tube plays an instructive role in establishing NC *Hox* expression, which is then modulated by permissive signals in the PA environments. Hence, the current model is that *Hox* expression initiated in the neural tube is not simply passively retained by migrating NC cells. The character-ization of essential sites bound by Hoxa2, Meis, and Pbx in the mouse *Hoxa2* and *Hoxb2* NC enhancers suggests that Hox and TALE-dependent auto-/cross-regulation may provide a mechan-ism for potentiating *Hox2* expression that is set up in the pre-migratory NC. Such auto-regulation has been shown for rhom-bomeric expression of many *Hox* genes[11,13,15] and may also be a general mechanism for pre-patterning the NC. Intriguingly, there appears to be context-dependent inputs that modulate the ability of Hox-response elements containing *Meis* and *Pbx-Hox* sites to potentiate activity in the hindbrain versus NC. For example, like the *Hoxb2* NC/r4 enhancer, *Hoxb1* has an auto-/cross-regulatory element dependent on *Meis* and *Pbx-Hox* sites, but it only mediates the expression in r4 and not in r4-derived NC[13]. Similarly, since *Hoxa2* is expressed in r2 but not in r2-derived NC[46], other regulatory mechanisms presumably prevent *Hox* expression in the r2-derived NC. This could include fibroblast growth factor signalling from the isthmus, which plays an important role in patterning PA1[47]. Further regulatory analyses will be required to elucidate the generality of Hox auto-/cross-regulation in NC.

The emergence of the NC during vertebrate evolution provides a key example of how regulatory codes coevolved with novel cell types in an animal body plan. Non-vertebrate deuterostomes, like the hemichordate *Saccoglossus* and the cephalochordate amphioxus, lack NC but deploy nested AP domains of *Hox* expression to pattern their nervous system[50–52]. This raises the intriguing possibility that the NC *Hox* code in ancestral vertebrates evolved from the transfer of a deuterostome neural *Hox* prepattern[14]. Alternatively, the NC *Hox* code may have arisen independently, by evolution of new regulatory inputs into *Hox* genes. A further possibility invokes a combination of both, with shared inputs creating a *Hox* prepattern and independent inputs evolving to modulate this in a tissue-specific manner. Our investigation of ancestral *Hox2* NC regulation in vertebrates sets the stage for examining the emergence of *Hox* regulation in NC during chordate evolution. This requires comparison of deuterostome development, with a focus on non-vertebrate deuterostome cell types that may be evolutionarily related to NC[1].

Studies in tunicates and cephalochordates suggest that they employ similar gene regulatory programs to specify the neural plate border[53–55]. Recent studies in tunicates, the vertebrate sister group (Fig. 8a), have identified certain embryonic cell populations that display some characteristics of NC cells. For example, in *Ecteinascidia turbinata*, a colonial tunicate, the trunk lateral cells originate beside the neural tube and migrate to give rise to pigmented cell types, leading to their designation as NC-like cells[56]. Trunk lateral cells are also identifiable in *Ciona*, where they express homologues of some key genes of the vertebrate NC-GRN, including *Tfap2α* and *Twist*[57]. However, the homology of trunk lateral cells to NC has been called into question[58].

*Ciona Hox* genes are dispersed across two chromosomes and residual spatial colinearity of expression in the nervous system has been detected for some of them[59]. This may be a general feature of tunicates, with similar *Hox* cluster disintegration seen in species from other tunicate classes[60,61]. Comparison with amphioxus, which has a single *Hox* cluster and nested colinear *Hox* expression along the AP axis of the neuroepithelium[51,52,62], suggests that tunicates are relatively divergent in terms of their *Hox* genomic content and expression. *Ciona intestinalis Hox2* expression has not been detected in the developing neural tube or neural plate border, but has been described in the larval ectodermal atrial primordia[63]. No defects in larval morphogenesis were detected upon morpholino-mediated knockdown of *Hox2* in *Ciona* embryos[63], so the roles of *Hox2* in tunicate embryonic patterning and its placement in tunicate developmental GRNs remain unclear.

In other non-vertebrate deuterostomes, the coincident expression of *Hox* and *Meis* genes in *Saccoglossus*[50,64] and amphioxus neuroectoderm[52,65] leads us to speculate that these factors may have comprised an ancestral deuterostome regulatory circuit involved in neuroectodermal patterning (Fig. 8a). Upon evolution of NC, pre-existing auto- and cross-regulatory interactions within this network may have served to maintain expression of these factors in the migrating NC. Investigation of regulatory interactions between *Hox* and TALE genes in invertebrate deuterostomes, combined with characterisation of the *cis*-regulatory elements involved, could help to address whether such Hox-TALE interactions were ancestral to deuterostomes and were employed in coupling *Hox* genes to NC during chordate evolution. Amphioxus lacks NC[54], but interspecies regulatory analysis, assaying activity of regions of the amphioxus Hox cluster in chicken and mouse embryos, revealed that some *cis*-elements are capable of mediating reporter expression in the hindbrain, placodes, and NC[66]. The activity of these elements in amphioxus is unknown but it will be important to investigate whether these represent ancestral neural elements with a capacity for mediating NC activity in vertebrates.

In summary, our finding of functionally conserved *Meis* and Pbx-*Hox* sites in lamprey and gnathostome *Hox2* NC enhancers focuses attention on the role of these factors in NC development. Meis and Pbx play important roles in patterning diverse tissues during development, including the hindbrain and NC, some of which may be independent of Hox[30–33]. For example, mouse embryos with a conditional deletion of *Meis2* in NC display abnormalities in patterning the bones and connective tissue in PA1, where *Hox* genes are not expressed[32]. Hence, they could also serve as cofactors for other transcription factors[34], or have independent roles in patterning NC. Therefore, while they have not been linked to the current NC-GRN, TALE factors (Pbx and Meis) may be important components in this network. If so, these transcription factors could be part of a mechanism that couples *Hox* genes to the NC GRN in vertebrate evolution. The conserved expression of *Meis* genes in NC from gnathostomes and lamprey is consistent with an ancestral role in NC and their roles and interactions in NC development require further study.

## Methods

**Sequence alignment**. Global sequence alignment of *Hox* genomic loci (Fig. 3a) was performed using Multi-LAGAN[28], with human as the baseline sequence and conserved sequences defined by 60% conservation over 40 bp. Sequence alignments of *Hox2* enhancers (Figs. 4f, 5a, b; Supplementary Figures 3, 5, 6) were performed using AlignX in VectorNTI (Life Technologies).

**Enhancer elements**. Enhancer elements were selected from the published data or based on sequence conservation in cross-species alignments. The DNA for each element was amplified by PCR from genomic DNA or from pre-existing plasmids using KOD Hot Start Master Mix (Novagen). The following primers were used for amplification. The sequences in uppercase represent homology to genomic DNA, and adaptor sequences for cloning are in lowercase text. References are given for primary literature in which the enhancers were identified.

*crestin−1 kb(zf)*[26],
F:5′-ccctcgaggtcgacGCTGAAATCTTGGGCATCTC-3′;
R: 5′-gaggatatcgagctcGCTGGGTTACTGAGGTGAC-3′;
*crestin−296 bp*[26],
F: 5′-agggtaatgagggccCCGCAGATGTTCTAGTACCC-3′;
R: 5′-gaggatatcgagctcgGGGTTAAAACAACACATTGATTAACCTGG-3′;
*Hoxa2(m)ΔNC3-1/ΔNC3-2*[10],
F: 5′-agggtaatgagggcccAGATCTGAATGCTGGAGC-3′;
R: 5′-tcgcccttcatagcctcgagGGTACCTTCTCTCCCTCAAAC-3′;
*hoxα2 elementA*,
F: 5′-agggtaatgagggccCCATCGACATGTAAACGTGGG-3′;
R: 5′-tcctacgtcactggcGAGTAAGCGAGGTCGTGG-3′.

The following enhancer elements were cloned into the Hugo's lamprey construct (HLC) reporter vector in a previous study focusing on the hindbrain:[18] *hoxa2b(zf)*, *Hoxa2a(f)*, *Hoxa2(m)*, and *hoxα2 −4 kb*.

**Generation of reporter constructs**. The HLC vector was created in a previous study[18]. PCR-amplified enhancer elements were purified using the QIAquick PCR Purification Kit (Qiagen) and cloned into HLC by Gibson Assembly using the Gibson Assembly Master Mix (NEB). The mouse *c-Fos* promoter was cloned into the *hoxα2 −4 kb*-HLC vector that had been linearized either by NcoI (for *hoxα2 −4 kb cfosV1*) or by AscI and NcoI (for *hoxα2 −4 kb cfosV2*). The following primer pairs were used to amplify the mouse *c-Fos* promoter from a plasmid template:
*hoxα2 −4 kb cfosV1*,
F: 5′-ctccgtcaaggcagcCCAGTGACGTAGGAAGTCCATC-3′;
R: 5′-ctcgcccttgctcaccatggTGGCGACCGGTGGATCCT-3′;
*hoxα2 −4 kb cfosV2*,
F: 5′-cgcctattggctgggCCAGTGACGTAGGAAGTCCATC-3′;
R: 5′-ctcgcccttgctcaccatggTGGCGACCGGTGGATCCT-3′.

Site-directed mutagenesis of enhancers was achieved by Gibson Assembly. For each mutation variant, two partially overlapping amplicons (left (L) and right (R)) containing the desired mutation were generated by PCR and then assembled into the linearized *HLC* vector by 3-fragment assembly. The following primers were used.
*crestin−296 bpΔSox10*,
L_F: 5′-agggtaatgagggccCCGCAGATGTTCTAGTACCC-3′;
L_R: 5′-ctagagatcgtcgcaTCTCTACGAAATTGTGCTTCTAGCAG-3′;
R_F: 5′-cgtagagatgcgacGATCTCTAGAAACATTAATGCATATGAACAAAAGC-3′;
R_R: 5′-gaggatatcgagctcgGGGTTAAAACAACACATTGATTAACCTGG-3′;
*crestin−296 bpΔTfap2α*,
L_F: 5′-agggtaatgagggccCCGCAGATGTTCTAGTACCC-3′;
L_R: 5′-ggatgtgcttaagttGAGCACATGACCAGGAGTC-3′;

*R_F:5′-atgtgctcaacttaaGCACATCCTGCTAGAAGCAC-3′*;
*R_R: 5′-gaggatatcgagctcgGGGTTAAAACAACACATTGATTAACCTGG-3′*;
*crestin−296 bpΔMyc,*
*L_F: 5′-agggtaatgagggccCCGCAGATGTTCTAGTACCC-3′*;
*L_R: 5′-aaggcgagtctagaACCAGGAGTCAATTAAAAGTCTCGTG-3′*;
*R_F:5′-ctcctggttctagaCTCGGCCTTGGGCACATCC-3′*;
*R_R: 5′-gaggatatcgagctcgGGGTTAAAACAACACATTGATTAACCTGG-3′*;
*hoxa2 −4 kb Δkrox20,*
*L_F: 5′-agggcccgggatcccTCGAGCCTGCAGGAAGCTTAAG-3′*;
*L_R: 5′-cccggtaaGCGGGACTCTGTTATCTCC-3′*;
*R_F: 5′-agtcccgcTTACCGGGATTCCGGCCAC-3′*;
*R_R: 5′-tctacgacgacgacgacgtcgaggTCGACGCAAAGAAGCCGG-3′*;
*hoxa2 −4 kb ΔNC3,*
*L_F: 5′-agggcccgggatcccTCGAGCCTGCAGGAAGCTTAAG-3′*
*L_R: 5′-ccctcgctCTTGCCCTGCACAAATACTCAG-3′*;
*R_F: 5′-agggcaagAGCGAGGGGCTCCGGAAAG-3′*;
*R_R: 5′-tctacgacgacgacgacgtcgaggtcgaCGCAAAGAAGCCGGCCCC-3′*.

**Zebrafish and lamprey experiments.** This study was conducted in accordance with the Guide for the Care and Use of Laboratory Animals of the National Institutes of Health and protocols were approved by the Institutional Animal Care and Use Committees of the Stowers Institute (zebrafish, RK Protocol #2015-0149 and Protocol #2018-0184) and California Institute of Technology (lamprey, Protocol #1436-17).

**Zebrafish reporter assay.** The wild-type Slusarski AB zebrafish line was used for embryo micro-injection experiments using Tol2-mediated transgenesis[67]. A standard injection mix containing 25 ng μl$^{-1}$ reporter plasmid (generated by miniprep), 35 ng μl$^{-1}$ Tol2 transposase messenger RNA, and 0.05% phenol red was micro-injected into one-celled embryos at an injection volume of 3–5 nl. Embryos were screened at 24–30 h post fertilization for fluorescent reporter expression using a Leica M205FA microscope. In assaying reporter constructs by transient transgenesis, for each injected construct the tissue-specific GFP expression domains were noted, along with the number and proportion of screened embryos exhibiting GFP expression in each of those domains. The empty HLC reporter vector (without an enhancer) directs weak mosaic GFP expression in multiple cell types including neurons and muscle cells (Supplementary Figure 7a). The following pre-existing transgenic reporter lines were used for this study: *Tg(dr.hoxa2b:eGFP)*, *Tg(fr. Hoxa2a:eGFP)*[12,18]. The line *Tg(mm.Hoxa2b:eGFP)* was generated in this study from a founder that had been micro-injected with *Hoxa2(m)-HLC*. Fluorescent and bright-field imaging were performed with Leica DFC360FX and DFC405C cameras and LAS AF imaging software. Images were cropped and altered for brightness and contrast using Adobe Photoshop CS5.1.

Zebrafish *crestin* reporter expression was analysed using *Tg2(−4.5_crestin: EGFP)* that uses the core long terminal repeat (*LTR*) elements of crestin located −4.5 kb upstream of the putative *crestin* open reading frame[26]. Transient assays for mutated transcription factor binding sites in lamprey were performed using the minimal 296 bp *crestin LTR* element with the previously reported mutations as tested in zebrafish[26].

**Lamprey reporter assay.** Lamprey transient transgenesis was performed using *P. marinus* embryos at the one-cell stage and I-SceI meganuclease-mediated transgenesis[18,68]. Injection mixes containing 20 ng μl$^{-1}$ reporter plasmid (generated by miniprep), 1× CutSmart buffer (NEB), and 0.5U μl$^{-1}$ I-SceI enzyme (NEB) in water were incubated at 37 °C for 30 min prior to micro-injection at a volume of ~2 nl per embryo. Embryos were screened for fluorescent reporter expression using a Zeiss SteREO Discovery V12 microscope. For each injected construct, the tissue-specific GFP expression domains were noted, along with the number and proportion of screened embryos exhibiting GFP expression in each of those domains. The empty HLC reporter vector (without an enhancer) directs GFP expression in ectoderm, yolk cells, as well as in cells dorsal to the yolk (Supplementary Figure 7b). Since transient reporter assays generate mosaic reporter expression patterns, variation in levels and domains of GFP expression are observed between embryos. For imaging we selected embryos with GFP-expressing patterns representative of the expression potential of the reporter construct, as inferred from screening more than 100 injected embryos. GFP-expressing embryos were imaged using a Zeiss SteREO Discovery V12 microscope and a Zeiss Axiocam MRm camera with AxioVision Rel 4.6 software. Images were cropped and altered for brightness using Adobe Photoshop CS5.1. Selected GFP-expressing embryos were fixed in MEMFA and dehydrated in methanol for in situ hybridisation.

**Cloning lamprey in situ hybridisation probes.** Probes were designed based on characterised or predicted gene sequences[22], amplified from *P. marinus* genomic DNA or st18–26 embryonic complementary DNA by PCR using KOD Hot Start Master Mix (Novagen) and cloned into the *pCR4-TOPO* vector (Life Technologies). The size of each amplified fragment is indicated (in bp). The following primers were used for PCR:
*hoxβ1* (674 bp, partial 3′-untranslated region),
*F: 5′-ATGCTCCCTCAACTCCATCC-3′*;
*R: 5′-TGACCTCTTTCTCGCATGTAAGA-3′*;

*hoxδ2* (585 bp, exonic),
*F: 5′-ACCTCTGCGCGACTCCTC-3′*;
*R: 5′-CCAGACCTCCTCCTCCTCT-3′*;
*meisC* (573 bp, exonic),
*F: 5′-CTTTGAGAAGTGCGAGCTGG-3′*;
*R: 5′-GAAAATGCCGCGCTTCTTCT-3′*.
*eGFP, hoxα2,* and *hoxα3* probe sequences were previously reported[18].

**Lamprey in situ hybridisation.** Digoxygenin-labelled probes were generated by standard methods and purified using the MEGAclear Transcription Clean-Up Kit (Ambion). Lamprey embryos were staged according to Tahara et al.[69]. Lamprey whole-mount in situ hybridisation was performed on MEMFA-fixed embryos following established protocols[70], with the following additions to the protocol:[71] methanol-stored embryos were transferred into ethanol and left overnight prior to rehydration; embryos were treated with 0.5% acetic anhydride in 0.1 M triethanolamine after proteinase K treatment. For imaging, embryos were cleared in benzyl alcohol:benzyl benzoate and mounted in Permount (Fisher Scientific).

For sectioning after in situ hybridisation, embryos were transferred into 30% sucrose in phosphate-buffered saline, embedded in O.C.T. Compound and sectioned to 10-μm-thick cryosections. Images were taken using a Zeiss Axiovert 200 microscope with AxioCam HRc camera and AxioVision Rel 4.8.2 software.

**ES cell culture.** ES cells were cultured in feeder-free conditions using N2B27 + 2i media supplemented with 2000U mL$^{-1}$ of ESGRO (Millipore) on a gelatinized plate. KH2 ES cells[72] with epitope-tagged Hoxb1 (3XFLAG-MYC) were used for Hoxb1 ChIP using anti-flag antibody (F1804-Sigma). Unmodified KH2 cell lines were used for ChIP experiments for Pbx (SC-888; Santa Cruz), Meis (SC-25412; Santa Cruz), Prep1 (ab55603; Abcam), Prep2 (sc-292315X; Santa Cruz) and EP300 (Sc-585X; Santa Cruz). Cells were differentiated to neuroectoderm in differentiation media containing DMEM + 10% (vol/vol) Serum + NEAA + 3 μM RA for a requisite length of time. Cells were harvested at 80–90% confluency.

**Chromatin immunoprecipitation-sequencing.** ChIP-seq was performed according to the Upstate protocol as described[73] with modifications. Cells were fixed with 1% formaldehyde by incubating at 37 °C for 11 min. The reaction was quenched for 5 min by the addition of 1/10th volume of 1.25 M glycine. Cells were sonicated for 25 min in a Bioruptor at high setting and 30 s on–off cycle. Respective antibodies attached to sepharose-A beads were used for immunoprecipitation. Sequencing of ChIP-seq libraries was performed on the Illumina HiSeq 2500, 51 bp single end. Raw reads were aligned to the UCSC mm10 mouse genome with bowtie2 2.2.0[74]. Primary reads from each bam were normalized to reads per million and bigWig tracks visualized at the UCSC genome browser (https://genome.ucsc.edu/).

**Assay for transposase-accessible chromatin-sequencing.** Assay for transposase-accessible chromatin-sequencing was performed as described previously[75]. Fifty thousand cells were counted using the sceptre 2.0 cell counter (EMD Millipore). The tagmentation reaction was performed using the Nextera DNA Library Prep Kit (Illumina, FC-121-1030) and libraries indexed using the Nextera Index Kit (Illumina, FC-121-1011). Libraries were size selected by Bluepippin (Sage Science) and sequenced on Illumina HiSeq. 2500 instrument. Following sequencing, Illumina Real Time Analysis v1.18.64 and CASAVA v1.8.2 were run to demultiplex reads and generate FASTQ files.

**Reporting summary.** Further information on experimental design is available in the Nature Research Reporting Summary linked to this article.

## Data availability
The authors declare that all data supporting the findings of this study are available within the article and its supplementary information files or from the corresponding author upon reasonable request. All raw sequencing data from this study underlying Fig. 7a have been deposited in the NCBI BioProject database [https://www.ncbi.nlm.nih.gov/bioproject] under accession code PRJNA341679 and Sequence Read Archive under accession code SRP079975 and PRJNA503882. Original data underlying this manuscript can be accessed from the Stowers Original Data Repository at [http://odr.stowers.org/websimr/]. A reporting summary for this Article is available as a Supplementary Information file.

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

## Acknowledgements

We thank Dorit Hockman, Tetsuto Miyashita, and Megan Martik for lamprey husbandry assistance, the Stowers Institute aquatics facility for zebrafish care, and Histology facility for sectioning assistance. This study was conducted in accordance with the recommendations in the Guide for the Care and Use of Laboratory Animals of the NIH and protocols approved by the Institutional Animal Care and Use Committees of the Stowers Institute (Zebrafish, RK Protocol: #2015-0149), California Institute of Technology (lamprey, MEB Protocol: #1436-11), and the veterinary office of UZH and the Canton of Zürich. K.D.P., C.H., and C.M. were supported by Science Foundation (SNSF) professorship (C.M. grant 170623), a Marie Curie Career Integration Grant from the European Commission (C.M. grant PCIG14-GA-2013-631984), the Swiss Cancer League, and the Canton of Zürich. H.J.P., B.D.K., L.M.W., and R.K. were supported by the Stowers Institute (R.K. grant #2013-1001). S.A.G. and M.E.B. were supported by grants R01NS086907 and R01DE017911.

## Author contributions

H.J.P., M.E.B. and R.K. conceived this research programme. H.J.P., B.D.K., K.D.P. and C. H. conducted the experiments. S.A.G. performed lamprey husbandry. C.K.K. and C.M. developed the *crestin* transgenic zebrafish line and associated constructs. H.J.P., B.D.K., C.M., L.M.W., M.E.B. and R.K. analysed the data, discussed the ideas and interpretations, and wrote the manuscript.

## Additional information

**Competing interests:** The authors declare no competing interests.

