## [Peer Review File · Nature Communications]

This manuscript has been previously reviewed at another journal that is not operating a transparent peer review scheme. This document only contains reviewer comments and rebuttal letters for versions considered at Nature Communications. Mentions of the other journal have been redacted.

Reviewers' Comments:

Reviewer #1:

Remarks to the Author:

The revised manuscript is now much improved, especially considering that the authors have removed much of the text that ran the risk of over-interpretation. After another careful reading, I do have some additional issues/concerns that I would like the authors to address. Much of this can be done by revising and/or adding text for clarification. I provide detailed comments below.

MAJOR CONCERNS

Use of Crestin enhancer element

I understand the general appeal of testing the zebrafish Crestin enhancer element in lamprey. The result is certainly interesting: as the authors point out, it seems likely that the transcriptional inputs that drive endogenous Crestin expression in the neural crest of zebrafish are the same ones activating the element in lamprey neural crest (e.g., SoxE, Myc, Tfap2a). Thus, lampreys can “decode” Crestin regulatory elements.

However, I just cannot see the logical connection between this particular experiment and the rest of the paper, which is about the ancestral role(s) of Hox2 regulation in neural crest cells. What relevance does the Crestin experiment have to this topic? The authors mention that they were looking for conservation of NC upstream of Hox2 activity, but they never show the connection between the supposed “upstream” activity of Crestin and the link between this and Hox2 activity. The authors go on to show that the gnathostome Hox2 neural crest elements can be read out in the neural crest of lamprey embryos. This is the relevant set of experiments. Where does Crestin fit in here? Yes, it marks neural crest, but the inputs described (SoxE, Tfap2a, Myc, etc.) have nothing to do with Hox2 regulatory activity in the neural crest, hence the lack of any logical connection. Overall, the inclusion of the Crestin data seems out of place and interrupts the natural progression of the paper. I see at least two possibilities: 1) remove the data altogether, or 2) if the authors really want to include the data, then put it in the supplementary file, remove most of the text (and the entire subheading) and simply mention in the section on Hox2 enhancers in the neural crest that other gnathostome enhancer elements additionally suggest conservation of regulatory control in neural crest (i.e., Crestin).

The benefit of 1) is that I think the Crestin expression could actually be pursued as a separate project. Crestin is known to be zebrafish-specific (teleost-specific?) and hence evolved relatively recently in that lineage. So, you've got a nice example of how a newly evolved gene could have become integrated into the neural crest GRN—ostensibly by the recruitment/evolution of combinatorial transcription factor binding sites of key neural crest factors (SoxE, Myc, Tfap2a). So, again, it's a nice result, just one that I can't see fitting into the current manuscript.

Lines 55-65: The authors have done a good job modifying much of the text to avoid over-interpretation, particularly in reference to Hox neural crest regulation being linked to a Hox prepatter in the nervous system of invertebrate deuterostomes. However, this paragraph still seems to prepare the reader for a test of this notion, which, again the data do support. The authors should substantially revise this paragraph and some text in the beginning of the following paragraph (e.g., Lines 66-67, 74) to better frame what their goal is and what they can test. Perhaps it could be re-framed to address the ancestral mechanisms of Hox2 neural crest regulation, which are largely unknown, etc. By making this change, it may be more feasible to just delete this paragraph altogether.

Inference of ancestral Hox2 regulation: Authors argue that lamprey *hoxa2* enhancer activity retains much of the ancestral vertebrate Hox2 regulatory activity. But, if *hoxa2* and *hoxδ2* in lamprey are cyclostome-specific duplications, then how is it justifiable to infer ancestral conditions in vertebrates from this lineage-specific duplication? An analogy here might be useful. Imagine trying to infer regulatory conditions operating in a gene present in the last common bony vertebrate ancestor by analyzing the regulatory activity of one of two paralogues resulting from a

teleost-specific duplication event. This is particularly confusing and problematic as presented in Fig. 9 in which the authors argue that one of the resulting Hox2 paralogues was lost in the cyclostome lineage and with the other being duplicated once more to give *hoxδ2* and *hoxα2*. But yet, they go on to use this single *hoxα2* enhancer to argue for ancestral, pre-duplicate conditions in the last common vertebrate ancestor. This comes across like a very circuitous route to go from regulatory control after lineage-specific duplications to conditions in the last common vertebrate ancestor.

Perhaps I've misunderstood something, but this seems to be the case in this manuscript. Part of this can be resolved by the authors adding text to explain much more carefully the hypothetical scenario for when duplications took place, in what lineages, and what gene products resulted, which were lost, retained, etc. as it relates to the Hox2 locus and comparable loci in cyclostomes. I realize that the authors have made some modifications to the text along similar lines in response to another Referee, but additional clarification couldn't hurt. But I would still like the authors to address the issue of inference of ancestral conditions as I've outlined above and explain why their approach is sound. Carefully articulated, this text should also be included in the manuscript.

MINOR CONCERNS

Line 50: "...this NC GRN does not (emphasis mine) include Hox genes." But then lines 52-54 go on to say that "It is unclear (emphasis mine) whether Hox networks...". These seem to be contradictory statements. Please revise for clarity and consistency.

Line 57: Please insert "the hemichordate" before *Saccoglossus* for consistency with "the cephalochordate amphioxus".

Line 210: Please change to "appear to be regulated" rather than "are regulated" given that some of this involves in silico analysis.

Line 230: Please remove "strongly".

Lines 296-300: It's also possible that selection maintains minimal transcription factor binding sites, with intervening sequence evolving neutrally.

Line 313: Please add text here noting that divergence has also occurred among lamprey *hoxα2* and *hoxδ2* regulatory activity, but perhaps in unique ways. This provides balance and avoids too much focus on gnathostomes.

End of Line 315: Please provide supporting references.

Line 328: Please change "are actually ancient paralogs" to "may be ancient paralogs".

End of Line 332: As per line 313 above, please add text about the conditions in extant cyclostomes as well.

Line 374: Please change "reveal" to "suggest".

Line 377: Please delete "as retained" to "which may be retained in part".

Line 383: This sentence seems problematic as it was not clear from the beginning of the manuscript what the relationship among paralogs is/was. What evidence is there that one paralog was lost in lamprey? This relates to my comment above regarding inference of ancestral conditions.

On the use of "AP-2": I'm fairly certain that the nomenclature is now "Tfap2α". Please update throughout.

Figures in general: For the fluorescent/BW images, please change the arrowheads to a color other than white to contrast better against the reporter expression.

Figure 9a: please add some text here indicating the cyclostome lineage/lamprey lineage also shows divergence in Hox2 enhancer function separately from gnathostomes. As is, it only shows what's happened in gnathostomes.

Reviewer #2:

Remarks to the Author:

The authors analyze and compare regulation of gnathosome *hoxa2/hoxb2* to that of lamprey *hoxa2* to identify shared and/or independent mechanisms driving axial patterning in the hindbrain and neural crest. This is a very sound analysis of regulation that indicates that the lamprey *hoxa2* regulation by Tale and Hox proteins in a single enhancer may be the ancestral state of Hox2 regulation. In other words, Hoxa2 regulation by two separate enhancer is the result of subfunctionalization of the single enhancer —

The authors responded very thoroughly to all my comments which were primarily technical. I only have a few points of clarification that will make the paper a bit easier to read through.

- 1) For the naive reader, in paragraph 2 of the introduction - it would be better to explicitly state that the Hox neuroectodermal prepatterning refers to the expression in neural tube- Or are you referring to the ancestral expression in invertebrate deuterostomes?
- 2) In Figure 1, for clarity it would help to have distances (kb) on the diagram or maybe the size of the enhancers would suffice. This comes back to what is the definition of a single enhancer- for example knowing what the size of NC3 is and the distance between NC3 and NC2 may be meaningful when interpreting the results.
- 3) In line 855, I would add (PA2) or (2) after r4-derived NC.
- 4) Figure 4, NC3 is deleted (lines 172-174) but the reason for that specific deletion over another is not explained until later in the paper and then in the discussion.
- 4) Line 163 -180 may be better placed after 183-186 after the global sequence alignment is discussed.
- 5) In figure 4b, the expression of any construct in lamprey rhombomeres is not evident or marked. However, in Supp 2b expression of *hox2b* expression in lamprey is quite clear. Do the authors truly believe that the gnathostome *hoxa2* enhancers drive expression in lamprey rhombomeres?

Reviewer #3:

Remarks to the Author:

The revised manuscript by Parker and colleagues, now under revision in Nature Communications, has significantly improved the previous version. I am glad to see that the authors found useful my suggestions, and, in those cases where they disagree or could not perform the suggested experiment, I also find their responses and rationale convincing and appropriate.

There is however an issue that I will further insist upon, and it is that of the 2R hypothesis (shared or independent). While I do agree with the authors that given the lack of definite evidence to support either scenario, showing the most parsimonious case is the most rational option (and thus I am not asking to make any changes on that regards), I find that the writing can still be a bit misleading.

- 1) L107-109: *Recent reconstructions based on comparisons of gene order at the chromosomal level between vertebrate species support a model in which the ancestor of cyclostomes and gnathostomes also had four Hox clusters (Sacerdot et al., 2018; Smith et al., 2018).*

Sacerdot and colleagues' comparison between the sea lamprey genome and their Amniota reconstructed genome gave a "clear majority of 1:4 patterns". However, that result can be explained not only by a shared 2R, but also by an independent 2R in each lineage. What Sacerdot et al. did is to mention that a shared 2R is the **most parsimonious scenario**, thus recognizing that it was not a direct evidence. From Sacerdot et al., 2018, page 11, related to their Fig. 7: "In addition, the clear 1:4 pattern is most parsimoniously explained if the Gnathostomes and the lamprey lineages share the 1R-2R duplications in their common ancestral history, which places the divergence of the Gnathostomes from the lamprey lineage after the 1R-2R duplications."

Surprisingly, Sacerdot et al. 2018 did not check whether the post-2R chromosomal fusion/fission events that they found were conserved with the sea lamprey genome, which in case of being identified in the latter would have been an irrefutable evidence for a shared 2R between gnathostomes and cyclostomes. Also, see Holland and Daza 2018 (Genome Biology, 19(209), 2–5. <http://doi.org/10.1186/s13059-018-1592-0>), from where I cite textually: "The study by Sacerdot and colleagues may be the best estimate yet of the history of chromosome duplication, fusion and fission in early vertebrate evolution, although a definitive answer as to whether the second WGD occurred at the base of the vertebrates or after the agnathan/gnathostome split may prove elusive. [...] The inclusion of more agnathan genomes (e.g. those from the Southern hemisphere) could also help distinguish chromosome-scale events that occurred before the agnathan/gnathostome split from those that occurred independently in agnathans. The fact that both lamprey and hagfish genomes appear to have six Hox clusters, indicates that they will not provide the final solution. It is unfortunate that these are the only extant agnathans and that time machines exist only in fiction. Therefore, the argument as to when the second WGD occurred may never be entirely settled."

Mentioning thus that Sacerdot et al.'s results "support" a model of 4 Hox clusters in the common ancestor of gnathostome and cyclostomes is too strong a statement, given the fact that their results are also compatible with a last common ancestor of vertebrates possessing 2 Hox clusters.

My suggestion is that the authors delete this sentence between L107-109, given that the whole paragraph would still keep its meaning unaltered, or just delete the reference to Sacerdot et al. here and leave just that of Smith et al., 2018 (who, as a side note, although recognized the presence of these ancestral 4 Hox clusters, notice that not necessarily due to a shared 2R)

2) Figure 9 legend: *'These scenarios assume that duplication events that gave rise to 4 Hox clusters in early vertebrates occurred prior to the cyclostome/gnathostome split, as suggested by molecular phylogenetic approaches (Kuraku et al., 2009), analysis of synteny patterns of duplicate genes (Smith et al., 2013), and comparison between chordate linkage groups and reconstructed chromosomes of the hypothetical amniote ancestor (Sacerdot et al., 2018). However, it is also possible that independent genome duplication events may have occurred in cyclostome and gnathostome lineages.'*

Molecular phylogenetic approaches are not appropriate to resolve the timing of the whole-genome duplication events, as the authors recognize themselves in L121-122 ("This may be due to the limitations of phylogenetic analyses in resolving the relative timing of ancient duplication events²⁵"). In this regards, Putnam et al., 2008 did a larger phylogenetic analysis and the results were inconclusive to resolve this question.

Sacerdot's comparison between the chordate linkage groups and the reconstructed amniote genome led them to suggest that at least the last common ancestor of amniotes, not the entire vertebrate group, had already undergone the 2R. They suggest, as I explain above, a shared 2R between gnathostomes and cyclostomes as the most parsimonious scenario when comparing the sea lamprey germline genome with their Amniote genome.

For these reasons and those explained previously, I suggest to change the legend to:

'These scenarios assume that duplication events that gave rise to 4 Hox clusters in early vertebrates occurred prior to the cyclostome/gnathostome split as the most parsimonious explanation (Smith et al., 2018; Sacerdot et al., 2018). However, it is also possible that independent genome duplication events may have occurred in cyclostome and gnathostome lineages (see Holland and Daza, 2018 for a recent discussion).'

Also, I have changed here Smith et al., 2013 to 2018, which I guess is the reference the authors wanted to cite. Please, correct if needed.

- Minor changes:

3) Lines 417 and 420: Correct nomenclature of genes never includes the abbreviation of the species in a gene's name. Please, change *ci-Hox2* for *Ciona intestinalis Hox2* in the first case. The second instance could be changed to "No defects in larval morphogenesis were detected upon morpholino-mediated knockdown of *Hox2* in *Ciona* embryos⁶⁶"

4) Figures 7 and 8 are related, have the authors considered putting them together into a single figure?

5) Supplementary Figure 2, legend. Change " Δ NC3_2 and Δ NC3_2" to " Δ NC3_1 and Δ NC3_2"

Response to reviewers' comments:

We thank the reviewers for their helpful suggestions, and we have modified the text and figures to add clarity and address their points. Below is a point by point list of the modifications and responses. For ease in reading, we have copied the reviewers comments in black text and noted our responses in blue.

REVIEWERS' COMMENTS:

Reviewer #1 (Remarks to the Author):

The revised manuscript is now much improved, especially considering that the authors have removed much of the text that ran the risk of over-interpretation. After another careful reading, I do have some additional issues/concerns that I would like the authors to address. Much of this can be done by revising and/or adding text for clarification. I provide detailed comments below.

MAJOR CONCERNS

Use of Crestin enhancer element

I understand the general appeal of testing the zebrafish Crestin enhancer element in lamprey. The result is certainly interesting: as the authors point out, it seems likely that the transcriptional inputs that drive endogenous Crestin expression in the neural crest of zebrafish are the same ones activating the element in lamprey neural crest (e.g., SoxE, Myc, Tfap2a). Thus, lampreys can “decode” Crestin regulatory elements.

However, I just cannot see the logical connection between this particular experiment and the rest of the paper, which is about the ancestral role(s) of Hox2 regulation in neural crest cells. What relevance does the Crestin experiment have to this topic? The authors mention that they were looking for conservation of NC upstream of Hox2 activity, but they never show the connection between the supposed “upstream” activity of Crestin and the link between this and Hox2 activity. The authors go on to show that the gnathostome Hox2 neural crest elements can be read out in the neural crest of lamprey embryos. This is the relevant set of experiments. Where does Crestin fit in here? Yes, it marks neural crest, but the inputs described (SoxE, Tfap2a, Myc, etc.) have nothing to do with Hox2 regulatory activity in the neural crest, hence the lack of any logical connection. Overall, the inclusion of the Crestin data seems out of place and interrupts the natural progression of the paper. I see at least two possibilities: 1) remove the data altogether, or 2) if the authors really want to include the data, then put it in the supplementary file, remove most of the text (and the entire subheading) and simply mention in the section on Hox2 enhancers in the neural crest that other gnathostome enhancer elements additionally suggest conservation of regulatory control in neural crest (i.e., Crestin).

The benefit of 1) is that I think the Crestin expression could actually be pursued as a separate project. Crestin is known to be zebrafish-specific (teleost-specific?) and hence evolved relatively recently in that lineage. So, you've got a nice example of how a newly evolved gene could have become integrated into the neural crest GRN—ostensibly by the recruitment/evolution of combinatorial transcription factor binding sites of key neural crest factors (SoxE, Myc, Tfap2a). So, again, it's a nice result, just one that I can't see fitting into the current manuscript.

We have modified this according to suggestion 2 of the reviewer. We eliminated the special section discussing crestin and combined it with analysis of the Hox2 NC enhancers. The data and text relating to the deletion mutants of transcription factor binding sites were removed from the main paper and placed in a supplementary file (Supplementary Fig.1).

Lines 55-65: The authors have done a good job modifying much of the text to avoid over-interpretation, particularly in reference to Hox neural crest regulation being linked to a Hox prepatter in the nervous system of invertebrate deuterostomes. However, this paragraph still seems to prepare the reader for a test of this notion, which, again the data do support. The authors should substantially revise this paragraph and some text in the beginning of the following paragraph (e.g., Lines 66-67, 74) to better frame what their goal is and what they can test. Perhaps it could be re-framed to address the ancestral mechanisms of Hox2 neural crest regulation, which are largely unknown, etc. By making this change, it may be more feasible to just delete this paragraph altogether.

We agree that this paragraph is out of place and have removed it from the introduction, incorporating part of it into the discussion. In its place we have written a couple of sentences to frame the goal of addressing ancestral mechanisms of Hox2 neural crest regulation, as suggested.

Inference of ancestral Hox2 regulation: Authors argue that lamprey *hoxα2* enhancer activity retains much of the ancestral vertebrate Hox2 regulatory activity. But, if *hoxα2* and *hoxδ2* in lamprey are cyclostome-specific duplications, then how is it justifiable to infer ancestral conditions in vertebrates from this lineage-specific duplication? An analogy here might be useful. Imagine trying to infer regulatory conditions operating in a gene present in the last common bony vertebrate ancestor by analyzing the regulatory activity of one of two paralogues resulting from a teleost-specific duplication event. This is particularly confusing and problematic as presented in Fig. 9 in which the authors argue that one of the resulting Hox2 paralogues was lost in the cyclostome lineage and with the other being duplicated once more to give *hoxδ2* and *hoxα2*. But yet, they go on to use this single *hoxα2* enhancer to argue for ancestral, pre-duplicate conditions in the last common vertebrate ancestor. This comes across like a very circuitous route to go from regulatory control after lineage-specific duplications to conditions in the last common vertebrate ancestor.

Perhaps I've misunderstood something, but this seems to be the case in this manuscript. Part of this can be resolved by the authors adding text to explain much more carefully the hypothetical scenario for when duplications took place, in what lineages, and what gene products resulted, which were lost, retained, etc. as it relates to the Hox2 locus and comparable loci in cyclostomes. I realize that the authors have made some modifications to the text along similar lines in response to another Referee, but additional clarification couldn't hurt. But I would still like the authors to address the issue of inference of ancestral conditions as I've outlined above and explain why their approach is sound. Carefully articulated, this text should also be included in the manuscript.

We understand the concern and we have removed much of the text referring to the hypothetical timing of duplications and their relationship to ancestral origin. In relation to this point, we have also modified original figure 9 (now Fig. 8) to focus on comparing expression and regulatory properties. We have also modified and put in text to clarify how we infer ancestral regulation as well as pointing to a caveat in the interpretation. The following text was inserted:

'Lamprey hoxa2 is the only lamprey hox2 paralogue expressed in PA2 NC (Fig.2b,c) and we uncovered the presence of homologous functional motifs (Meis, Pbx, and Hox) in an upstream enhancer with activity in r4 and NC. Based on sequence conservation, position and regulatory activity, we infer that this enhancer is homologous to gnathostome Hoxa2/Hoxb2 NC enhancers. Sequence comparisons suggest that lamprey hoxδ2 has diverged and is missing these NC motifs but retains Krox20 and Sox sites, consistent with its expression in r3/r5. Since the lamprey hoxa2 NC enhancer exhibits the combined activity of the Hoxa2 and Hoxb2 enhancers, we consider it likely that it reflects the ancestral state. Thus, we suggest that Hox2 was ancestrally regulated in r4 and NC by a shared enhancer through inputs by Meis, Pbx, and Hox (Fig.8b). Alternatively, if the ancestral NC enhancer did not have r4 activity, then gnathostome Hoxb2 and the lamprey hoxa2 NC enhancers independently evolved the ability to mediate expression in r4.'

MINOR CONCERNS

Line 50: "...this NC GRN does not (emphasis mine) include Hox genes." But then lines 52-54 go on to say that "It is unclear (emphasis mine) whether Hox networks...". These seem to be contradictory statements. Please revise for clarity and consistency.

We agree that these statements seem contradictory as written. We have modified the first statement to reflect that the current model of the NC GRN does not yet include *Hox* genes, implying that *Hox* genes may well be part of the network:

"However, Hox genes have not yet been integrated within the current formulation of the NC GRN. This is in part because the mechanisms regulating Hox expression in NC are relatively unclear compared to current knowledge of Hox regulation in hindbrain segmentation."

Line 57: Please insert "the hemichordate" before *Saccoglossus* for consistency with "the cephalochordate amphioxus".

We have made this insertion and this text has been moved to the discussion.

Line 210: Please change to "appear to be regulated" rather than "are regulated" given that some of this involves in silico analysis.

We have made this change.

Line 230: Please remove "strongly".

We have removed "strongly" and inserted "may" to soften this statement:

'The lamprey hoxa2 NC enhancer exhibits the combined activity of both mouse Hoxa2 and Hoxb2 enhancers, suggesting that it may reflect the ancestral state.'

Lines 296-300: It's also possible that selection maintains minimal transcription factor binding sites, with intervening sequence evolving neutrally.

We have modified the relevant text to include the possibility that the sequence differences observed between the different enhancers outside of the conserved motifs are the result of neutral sequence evolution.

'While these differences in neighbouring sequences may have arisen by sequence drift and be functionally neutral, an intriguing alternative is that they may have functional significance in modulating binding to the conserved motifs and impacting r4 activity.'

Line 313: Please add text here noting that divergence has also occurred among lamprey *hoxa2* and *hoxδ2* regulatory activity, but perhaps in unique ways. This provides balance and avoids too much focus on gnathostomes.

This is a great suggestion, we have modified the text to note the divergence between the lamprey *Hox2* paralogues.

'The lamprey hoxa2 NC enhancer appears to have retained ancestral activity in both NC and hindbrain, while the paralogous Hoxa2 and Hoxb2 NC enhancers have differentially partitioned NC and hindbrain activities in the gnathostome lineage. Regulatory divergence has also occurred between lamprey Hox2 paralogues, with hoxδ2 appearing to have lost the ancestral sites for r4/NC enhancer activity.'

End of Line 315: Please provide supporting references.

We have added references to support this statement.

"TALE factors are part of an ancient patterning system(Hudry, Thomas-Chollier et al. 2014, Merabet and Mann 2016) that may have multiple roles in coupling Hox expression to the core NC GRN."

Line 328: Please change "are actually ancient paralogs" to "may be ancient paralogs".

We have removed the term 'paralogous' and modified the sentence as follows:

'Our analyses here resolve this paradox, providing evidence that the Hoxa2/Hoxb2 NC enhancers each retain conserved Meis, Pbx, and Hox binding sites, which are deployed in slightly different ways in mediating tissue-specific activities.'

End of Line 332: As per line 313 above, please add text about the conditions in extant cyclostomes as well.

We have made this change by adding the following sentence:

'Sequence comparisons suggest that lamprey hoxδ2 has diverged and is missing these NC motifs but retains Krox20 and Sox sites, consistent with its expression in r3/r5.'

Line 374: Please change "reveal" to "suggest".

We have made this change.

Line 377: Please delete "as retained" to "which may be retained in part".

We have made this change.

Line 383: This sentence seems problematic as it was not clear from the beginning of the manuscript

what the relationship among paralogs is/was. What evidence is there that one paralog was lost in lamprey? This relates to my comment above regarding inference of ancestral conditions.

We agree that the relationships among paralogues is not clear and we have deleted this sentence and modified Figure 9 (now Fig.8), as described above in dealing with lineage relationships and ancestral conditions.

On the use of “AP-2”: I’m fairly certain that the nomenclature is now “Tfap2 α ”. Please update throughout.

We have altered this name to *Tfap2 α* as suggested.

Figures in general: For the fluorescent/BW images, please change the arrowheads to a color other than white to contrast better against the reporter expression.

Thank you for this suggestion. We have changed the arrowheads in all of the BW images to a different colour for improved contrast against the white reporter signal.

Figure 9a: please add some text here indicating the cyclostome lineage/lamprey lineage also shows divergence in Hox2 enhancer function separately from gnathostomes. As is, it only shows what’s happened in gnathostomes.

We agree that it is important to refer to the regulatory divergence of *Hox2* paralogues in cyclostomes in this summary figure. We have added text to the figure (now Fig.8a) adjacent to the cyclostome branch of the phylogeny to indicate divergence of *Hox2* enhancer function in this lineage.

--

Reviewer #2 (Remarks to the Author):

The authors analyze and compare regulation of gnathosome *hoxa2/hoxb2* to that of lamprey *hox α 2* to identify shared and/or independent mechanisms driving axial patterning in the hindbrain and neural crest. This is a very sound analysis of regulation that indicates that the lamprey *hox α 2* regulation by Tale and Hox proteins in a single enhancer may be the ancestral state of Hox2 regulation. In other words, Hoxa2 regulation by two separate enhancer is the result of subfunctionalization of the single enhancer

The authors responded very thoroughly to all my comments which were primarily technical. I only have a few points of clarification that will make the paper a bit easier to read through.

1) For the naive reader, in paragraph 2 of the introduction - it would be better to explicitly state that the Hox neuroectodermal prepatter refers to the expression in neural tube- Or are you referring to the ancestral expression in invertebrate deuterostomes?

We have altered to the text to clarify as follows:

'This raises the intriguing possibility that the NC Hox-code in ancestral vertebrates evolved from the transfer of a deuterostome neural Hox prepatter'

2) In Figure 1, for clarity it would help to have distances (kb) on the diagram or maybe the size of the enhancers would suffice. This comes back to what is the definition of a single enhancer- for example knowing what the size of NC3 is and the distance between NC3 and NC2 may be meaningful when interpreting the results.

We have altered Figure 1 by writing the size (in bp) next to each of the enhancers as suggested. We note that the boxes representing individual elements within the enhancers are not drawn to scale, but the multiple sequence alignments in Supplementary Fig. 5 and 6 give precise bp lengths of each motif/element.

3) In line 855, I would add (PA2) or (2) after r4-derived NC.

We have added to the text in the figure legend to clarify the labelling of rhombomeres and pharyngeal arches in the schematic dorsal views, and to specifically note that r4-derived NC is in PA2, as follows:

'Enhancers are marked as black lines below the loci, with their activity domains illustrated by blue shading in schematic dorsal views of the hindbrain (r2-5) and pharyngeal arches (2-3). For the mouse loci, characterised cis-elements contributing to enhancer function are depicted as coloured boxes: hindbrain elements in purple and NC elements in green. Known direct inputs from transcription factors into these cis-elements are depicted by arrows, with unknown inputs shown as question marks. Hoxa2 is regulated in r4 and r4-derived NC (PA2) by independent enhancers (a).'

4) Figure 4, NC3 is deleted (lines 172-174) but the reason for that specific deletion over another is not explained until later in the paper and then in the discussion.

We have added the following to introduce NC3 and our rationale for making this deletion rather than another:

"In gnathostomes, NC Hoxa2 expression is regulated by 5' flanking elements (NC1-5) that partially overlap those of a separate r3/r5 enhancer (RE1-5, Krox20, Sox) (Fig.1a; Fig.4a)^{10,26}. Of these, the NC3 element is the most highly conserved: global sequence alignment using Multi-LAGAN²⁷ identified sequence conservation of NC3 extending to sharks (Fig.4a). In previous work, two 15bp deletions within NC3 were each found to abolish NC reporter expression in mouse¹⁰. To determine whether the same cis-elements are required for NC activity of Hoxa2(m) in lamprey, we generated two variants with these deletions in NC3."

5) Line 163 -180 may be better placed after 183-186 after the global sequence alignment is discussed.

We prefer to keep this paragraph in its current location because we have modified the section before it, on *crestin*. Hence, this is an important lead-in to the *Hoxa2* cross-species enhancer experiments.

6) In figure 4b, the expression of any construct in lamprey rhombomeres is not evident or marked. However, in Supp 2b expression of *hox2b* expression in lamprey is quite clear. Do the authors truly believe that the gnathostome *hoxa2* enhancers drive expression in lamprey rhombomeres?

The hindbrain expression driven by the *hoxa2* enhancers in lamprey is less marked than the expression in neural crest in these images and it is also weaker than the hindbrain expression driven by the *hoxb2a(zf)* enhancer in lamprey. However, we have validated that these *hoxa2* constructs do drive GFP expression in the lamprey hindbrain, but this expression is at lower levels, or the signal is less detectable, than the neural crest expression. These findings are in line with our previous analysis of rhombomeric enhancers in lamprey which showed that *hoxa2* generated the weakest domains of segmental expression while *hoxb2a* gave robust segmental patterns (Parker et al. Nature 2014). We also note that *hoxa2b(zf)* drove expression in the hindbrain at a lower frequency than the *hoxa2a(f)* and *Hoxa2(m)* constructs in lamprey, as seen in the injection statistics in Supplementary Table 2.

--

Reviewer #3 (Remarks to the Author):

The revised manuscript by Parker and colleagues, now under revision in Nature Communications, has significantly improved the previous version. I am glad to see that the authors found useful my suggestions, and, in those cases where they disagree or could not perform the suggested experiment, I also find their responses and rationale convincing and appropriate.

There is however an issue that I will further insist upon, and it is that of the 2R hypothesis (shared or independent). While I do agree with the authors that given the lack of definite evidence to support either scenario, showing the most parsimonious case is the most rational option (and thus I am not asking to make any changes on that regards), I find that the writing can still be a bit misleading.

1) L107-109: *'Recent reconstructions based on comparisons of gene order at the chromosomal level between vertebrate species support a model in which the ancestor of cyclostomes and gnathostomes also had four Hox clusters (Sacerdot et al., 2018; Smith et al., 2018).'*

Sacerdot and colleagues' comparison between the sea lamprey genome and their Amniota reconstructed genome gave a "clear majority of 1:4 patterns". However, that result can be explained not only by a shared 2R, but also by an independent 2R in each lineage. What Sacerdot et al. did is to mention that a shared 2R is the **most parsimonious scenario**, thus recognizing that it was not a direct evidence. From Sacerdot et al., 2018, page 11, related to their Fig. 7: "In addition, the clear 1:4 pattern is most parsimoniously explained if the Gnathostomes and the lamprey lineages share the 1R-2R duplications in their common ancestral history, which places the divergence of the Gnathostomes from the lamprey lineage after the 1R-2R duplications."

Surprisingly, Sacerdot et al. 2018 did not check whether the post-2R chromosomal fusion/fission events that they found were conserved with the sea lamprey genome, which in case of being identified in the latter would have been an irrefutable evidence for a shared 2R between gnathostomes and cyclostomes. Also, see Holland and Daza 2018 (Genome Biology, 19(209), 2–5. <http://doi.org/10.1186/s13059-018-1592-0>), from where I cite textually: "The study by Sacerdot and colleagues may be the best estimate yet of the history of chromosome duplication, fusion and fission in early vertebrate evolution, although a definitive answer as to whether the second WGD occurred at the base of the vertebrates or after the agnathan/gnathostome split may prove elusive. [...] The inclusion of more agnathan genomes (e.g. those from the Southern hemisphere) could also help distinguish chromosome-scale events that occurred before the agnathan/gnathostome split from those that occurred independently in agnathans. The fact that both lamprey and hagfish genomes appear to have six Hox clusters, indicates that they will not provide the final solution. It is unfortunate that these are the only extant agnathans and that time machines exist only in fiction. Therefore, the argument as to when the second WGD occurred may never be entirely settled."

Mentioning thus that Sacerdot et al.'s results "support" a model of 4 Hox clusters in the common ancestor of gnathostome and cyclostomes is too strong a statement, given the fact that their results are also compatible with a last common ancestor of vertebrates possessing 2 Hox clusters.

My suggestion is that the authors delete this sentence between L107-109, given that the whole paragraph would still keep its meaning unaltered, or just delete the reference to Sacerdot et al. here and leave just that of Smith et al., 2018 (who, as a side note, although recognized the presence of these ancestral 4 Hox clusters, notice that not necessarily due to a shared 2R)

We agree that the data of Sacerdot et al. is consistent with the model of a shared 2R but that, since this is based on parsimony, it is not direct evidence for a shared 2R, so our statement is too strong. Thus, we have changed the text as suggested by removing the reference to Sacerdot et al., leaving that of Smith et al.

2) Figure 9 legend: *‘These scenarios assume that duplication events that gave rise to 4 Hox clusters in early vertebrates occurred prior to the cyclostome/gnathostome split, as suggested by molecular phylogenetic approaches(Kuraku et al., 2009), analysis of synteny patterns of duplicate genes(Smith et al., 2013), and comparison between chordate linkage groups and reconstructed chromosomes of the hypothetical amniote ancestor (Sacerdot et al., 2018). However, it is also possible that independent genome duplication events may have occurred in cyclostome and gnathostome lineages.’*

Molecular phylogenetic approaches are not appropriate to resolve the timing of the whole-genome duplication events, as the authors recognize themselves in L121-122 (“This may be due to the limitations of phylogenetic analyses in resolving the relative timing of ancient duplication events²⁵”). In this regards, Putnam et al., 2008 did a larger phylogenetic analysis and the results were inconclusive to resolve this question.

Sacerdot’s comparison between the chordate linkage groups and the reconstructed amniote genome led them to suggest that at least the last common ancestor of amniotes, not the entire vertebrate group, had already undergone the 2R. They suggest, as I explain above, a shared 2R between gnathostomes and cyclostomes as the most parsimonious scenario when comparing the sea lamprey germline genome with their Amniote genome.

For these reasons and those explained previously, I suggest to change the legend to:

‘These scenarios assume that duplication events that gave rise to 4 Hox clusters in early vertebrates occurred prior to the cyclostome/gnathostome split as the most parsimonious explanation (Smith et al., 2018; Sacerdot et al., 2018). However, it is also possible that independent genome duplication events may have occurred in cyclostome and gnathostome lineages (see Holland and Daza, 2018 for a recent discussion).’

Also, I have changed here Smith et al., 2013 to 2018, which I guess is the reference the authors wanted to cite. Please, correct if needed.

We have made the changes to the legend as suggested.

- Minor changes:

3) Lines 417 and 420: Correct nomenclature of genes never includes the abbreviation of the species in a gene's name. Please, change *ci-Hox2* for *Ciona intestinalis Hox2* in the first case. The second instance could be changed to "No defects in larval morphogenesis were detected upon morpholino-mediated knockdown of *Hox2* in *Ciona* embryos⁶⁶"

We have made the changes to the text as suggested.

4) Figures 7 and 8 are related, have the authors considered putting them together into a single figure?

We prefer to keep them as separate figures so that they follow the order of the main text.

5) Supplementary Figure 2, legend. Change " Δ NC3_2 and Δ NC3_2" to " Δ NC3_1 and Δ NC3_2"

We have corrected this mistake as suggested.